# Estimates of Atlantic meridional heat transport from spatiotemporal fusion of Argo, altimetry and gravimetry data

Francisco M. Calafat<sup>1,2</sup>, Parvathi Vallivattathillam<sup>2</sup>, Eleanor Frajka-Williams<sup>3</sup>

<sup>1</sup>Physics Department, University of the Balearic Islands, Palma, 07122, Spain

<sup>2</sup>Marine Physics and Ocean Climate, National Oceanography Centre, Liverpool, L3 5DA, UK

<sup>3</sup>Institute of Oceanography, University of Hamburg, 20146, Germany

Correspondence to: Francisco M. Calafat (francisco.mcalafat@uib.es)

**Abstract.** Understanding how changes in Atlantic meridional heat transport (MHT) and the Earth's climate relate to one another is crucial to our ability to predict the future climate response to anthropogenic forcing. Attaining this understanding requires continuous and accurate records of MHT across the whole Atlantic. While such records can be obtained through direct ocean observing systems, these systems are expensive to install and maintain and thus, in practice, records of MHT derived in this way are restricted to a few latitudes. An alternative approach, based on hydrographic and satellite components of the global ocean observing system, consists of inferring heat transport convergence as a residual from the difference between ocean heat content (OHC) changes and surface heat flux. In its simplest form, this approach derives the OHC from hydrographic observations alone, however these observations are spatially sparse and unevenly distributed, which can introduce significant errors and biases into the MHT estimates. Here, we combine data from hydrography, satellite altimetry and satellite gravimetry through joint spatiotemporal modelling to generate probabilistic estimates of MHT for the period 2004-2020 at 3-month resolution across 12 latitudinal sections of the Atlantic Ocean between 65° N and 35° S. Our approach leverages the higher spatial sampling of the satellite observations to compensate for the sparseness and irregular distribution of the hydrographic data, leading to significantly improved estimates of MHT compared to those derived from hydrographic data alone. The fusion of the various data sets is done using rigorous Bayesian statistical methods that account for the spatial resolution mismatch between data sets and ensure an adequate representation and propagation of uncertainty. Our estimates of MHT at 26° N agree remarkably well with estimates based on direct ocean observations from the RAPID array, both in magnitude and phase of the variability, with a correlation of 0.68 for quarterly (3-monthly) time series and 0.81 after applying a 4-quarter running mean. For the period 2004-2017, the correlation increases to 0.78 and 0.92, respectively. The time-mean MHT at 26° N is also captured by our approach, with a value of 1.14 PW [1.01,1.27] (5-95% credible interval). Estimates of MHT at other latitudes are

also consistent with what we expect based on earlier estimates as well as on our current understanding of MHT in the Atlantic Ocean.

#### 1 Introduction

Changes of the Earth's climate since the Industrial Revolution are primarily a result of the excess heat trapped in the climate system by the accumulation of greenhouse gases, leading to global warming. A substantial portion of this extra heat, over 90%, has been absorbed by the world's oceans (Meyssignac et al., 2019; von Schuckmann et al., 2020), temporarily slowing the warming of the atmosphere, albeit at the cost of higher sea levels, accelerated ice sheet melting, and harm to marine ecosystems. However, the ocean's role extends beyond being merely a passive thermal buffer against global warming. It also plays a central role in mediating the climate system response to greenhouse gas emissions by helping to shape – through heat redistribution – the pattern of sea-surface warming, on which climate feedback processes depend (Andrews et al., 2018; Dong et al., 2020). Ocean currents also play a crucial part in regulating the regional climate by transporting heat poleward from the tropics and then releasing it into the atmosphere at higher latitudes (Woollings et al., 2012; Buckley and Marshall, 2016; Zhang et al., 2019; Yin and Zhao, 2021). In the Atlantic Ocean, heat is transported northward throughout the basin by a vast system of ocean currents – the Atlantic meridional overturning circulation (AMOC) (Frajka-Williams et al., 2019) – carrying warm surface waters northward and cold waters southward at deeper levels. Importantly, the AMOC is capable of storing heat (and carbon dioxide) deep in the ocean where it can remain sequestered for centuries before resurfacing thousands of kilometers away. This unique capability endows the AMOC with the potential to affect the global climate over long time scales.

Extensive research efforts in recent decades have been dedicated to monitoring the AMOC and meridional heat transport (MHT) through various multi-observational approaches (Frajka-Williams et al., 2019; Li et al., 2021), leading to significant progress in our understanding of these two critical climate-relevant factors (Srokosz et al., 2021). Nevertheless, despite this progress, significant knowledge gaps persist, such as questions about the latitudinal coherence of the AMOC (and MHT) and whether it is weakening (Jackson et al., 2022; Piecuch & Beal, 2023; Volkov et al., 2024), among other major concerns. Filling these remaining gaps is crucial to advancing our understanding of future climate change. However, ongoing efforts to achieve this are faced with challenges related to limitations in observing capability as well as in the methods currently being used for combining noisy and sparse data from multiple sources. This study is particularly motivated by those limitations and focuses, specifically in the context of quantifying MHT.

Past changes in Atlantic MHT have been estimated mainly through two different approaches. The first approach, employed by both the RAPID/Meridional Overturning Circulation and Heat-flux Array/Western Boundary Time Series (hereafter RAPID) programme (Cunningham et al., 2007; Johns et al., 2011; McCarthy et al., 2015, Johns et al., 2023) and the Overturning in the Subpolar North Atlantic Program (OSNAP) (Lozier et al., 2019; Li et al., 2021), calculates MHT directly by integrating the product of the temperature and cross-sectional velocity across designated transbasin sections (26°N in RAPID and 50°N-60°N in OSNAP). Temperatures and velocities are estimated based on hydrographic data from transport mooring arrays, Argo profiling floats and, in the case of the ONSAP section, also ocean gliders. Heat transport through the Florida Straits in the RAPID section is estimated based on measurements from a submarine cable (Volkov et al. 2024). This approach is widely regarded as the gold standard for monitoring both MHT and the AMOC, but it still comes with limitations. In particular, such an observing array system is time-consuming and expensive to install and maintain, making it impractical for ocean-wide monitoring. Consequently, estimates of MHT based on this approach are restricted to these two latitudes, and thus they are not sufficiently latitudinally dense to characterize the spatiotemporal structure of the MHT.

The second approach, which is the focus of this study, attempts to fill these existing latitudinal gaps in observing. It consists of inferring ocean heat transport convergence (HTC) as a residual from the imbalance between changes in ocean heat content (OHC) and surface heat flux (HF). Here, we shall concern ourselves mainly with the estimation of OHC changes and will rely on state-of-the-art estimates of HF derived elsewhere in the literature. However, it is important to emphasize that both components of the energy balance are crucial to the success of this approach in their own right, requiring their own thorough consideration. Most past studies derive OHC changes from gridded temperature (T) and salinity (S) data sets produced through objective analysis of hydrographic profiles from various instruments (e.g., Argo floats, bathythermographs, bottles, etc.) (Roberts et al, 2017; Cheng et al., 2020; von Schuckmann et al., 2020). While such profiles have almost ocean-wide coverage, they are spatially sparse (including the Argo era), almost non-existent below 2000 m, irregularly spaced, noisy and highly heterogenous across instruments in terms of accuracy. These data issues can introduce significant biases and uncertainties into the gridded T/S products and, by propagation, into the estimates of OHC changes, especially on regional scales, restricting the accuracy with which we can estimate MHT through this heat-budget approach. Some studies use T and S data from ocean reanalyses as an alternative to observations (Trenberth and Fasullo, 2017; Trenberth et al., 2019), but such reanalyses have biases and uncertainties of their own and crucially depend on the availability of hydrographic data for assimilation, thus they face similar issues to in-situ observations.

A promising solution to the issues discussed above, with a view to improving the accuracy of OHC estimates, is to combine hydrography-derived thermosteric (TS) and halosteric (HS) height anomalies with sea level (SL) from satellite altimetry and ocean mass (OM) from satellite gravimetry, leveraging the relatively good spatial sampling of the satellite observations. The key idea is to exploit the fact that TS is directly linked to OHC and that SL is related to TS through SL=TS+HS+OM. A special case of this approach arises when the focus is on estimating global average OHC because, on global scales, halosteric effects are negligible (Lowe and Gregory, 2006; Gregory et al., 2019) and thus TS (and OHC) can be derived from satellite data alone as the residual of SL and OM (Dieng et al., 2015; Meyssignac et al., 2019; Hakuba et al., 2021; Marti et al., 2022). However, the estimation of MHT requires knowledge of regional OHC changes and, on such scales, halosteric effects can no longer be ignored (Maes, 1998; Wang et al., 2017) and need to be estimated from hydrographic data. In this case, a straightforward combination of the satellite and hydrographic data is direct pointwise subtraction, where the operation TS=SL-OM-HS is performed independently at each grid point. This simple data-merging approach has already shown improvements over estimates of MHT based solely on hydrographic data (Meyssignac et al.; 2024), but it lacks a formal statistical framework to address the complexities of the three data sets. These complexities include data error structures, spatial dependencies and resolution mismatches, among others. For example, the three data sets differ in spatial resolutions, making them inherently incompatible without adjustments – a challenge known as the change of support problem (Gelfand et al., 2001; Gotway and Young, 2002). Moreover, each data set has unique and complex error structures in both time and space. Successfully integrating the three data sets requires accounting for both the resolution mismatch and the error structures within a statistically coherent framework that models all data sets and their associated uncertainties simultaneously and comprehensively.




One of the first attempts to quantify MHT through joint modelling of hydrographic and satellite data was presented in the work of Kelly et al. (2014, 2016). They estimated HTCs by evaluating the heat budget over latitudinally-bounded regions of the Atlantic Ocean based on data from hydrography, altimetry and gravimetry together with reanalysis-derived surface heat fluxes. They used a two-step procedure wherein they first calculated spatial averages over each of the regions independently for each data set and then assessed the heat budget through joint modelling of the spatially averaged time series. While conceptually simple, this procedure has several limitations. By first calculating spatial averages separately for each variable, the procedure ignores any spatial dependencies between the variables and loses the opportunity to leverage cross-variable spatial information, both of which can lead to suboptimal estimates of spatially averaged values. Also, such a modelling choice makes the estimation of uncertainties in the spatially averaged values challenging, often requiring ad-hoc or approximate methods.

Here, we present a Bayesian hierarchical framework (see Cressie & Wikle (2011) for a general description of spatiotemporal hierarchical models) for estimating MHT that combines data from hydrography, altimetry and gravimetry in a statistically rigorous way. Our approach extends that of Kelly et al. (2016) by accounting for spatiotemporal dependencies between processes (i.e., TS, HS, and OM) and enabling information sharing across the various data sets. This is achieved by simultaneous spatiotemporal modelling of the observational fields and their error structures, in contrast to time series modelling of spatially averaged values as done in Kelly et al. (2016). The idea of combining multi-source climatic data through spatiotemporal Bayesian modelling has been successfully used before, for example to assess Antarctic ice mass changes (Zammit-Mangion et al., 2014; Zammit-Mangion et al., 2015) and sea-level trends (Piecuch et al., 2018; Calafat et al., 2022), but to our knowledge has not been applied to MHT and merging hydrography, altimetry and gravimetry. Our approach overcomes the limitations of hydrography-only based analyses and addresses the issues associated with combining data from disparate sources, leading to more robust and accurate estimates of both OHC and MHT. Importantly, by considering error structures jointly, the hierarchical approach provides a coherent way to propagate uncertainty in the data, model and parameters through the analysis to the estimates of the MHT. We use our Bayesian hierarchical model (BHM) to produce observation-based probabilistic estimates of non-seasonal quarterly (3-month-averaged) MHT for the period 2004-2020 across 12 latitudinal sections over the Atlantic Ocean between 65°N and 35°S. The sections are shown in Fig. (1). Some of the sections have been chosen arbitrarily to cover most of the Atlantic Ocean, while others have been selected because MHT or AMOC volume transport estimates from direct ocean observations are available or will be in the future.




**Figure 1.** Latitude lines across which MHT is estimated using a spatiotemporal BHM. The coloured areas denote the regions over which the heat budgets are evaluated. Both the MHT across each latitude and the HTC in each budget region have been labelled with numbers and this is the notation that we follow in Section 5.1.

#### 140 **2 Data**



## 2.1. Hydrography-derived TS and HS heights

TS and HS heights are calculated using monthly gridded fields of T and S from the ISAS20 product (Gaillard et al., 2016; available at https://www.seanoe.org/data/00412/52367/), which provides data on a 1/2° × 1/2° grid for the period January 2002 to December 2020 and is based solely on Argo profiles. ISAS20 also provides uncertainty estimates for the objectively analyzed T and S fields. While Argo floats can go as deep as 2000 m, only about 58% of the Atlantic profiles have data below 1900 m on average over the period 2004-2020 (lower than that in the early years of that period and higher in the most recent years). In contrast, about 72% of the profiles reach, on average, a depth of at least 1500 m (see also Wong et al., 2020). For this reason, we decide to use only T and S data from the surface to 1500 m in the calculation of TS and HS (the contribution from below 1500 m is accounted for by inflating the uncertainty in the TS and HS data, as explained later). It is also important to mention that we exclude the Gulf of Mexico and the Caribbean Sea in the evaluation of the heat budgets (Fig. 1) as there appears to be a

problem with the hydrographic data from ISAS20 in those regions. We have tested the impact of excluding data in these regions on our estimates of MHT and found it to be minimal.

TS and HS changes reflect the expansion and contraction of the water column induced by T and S variations, respectively. Assuming that such variations in T and S are small relative to the time-mean value, TS and HS anomalies can be calculated at each horizontal grid point (latitude – longitude) and for each month as follows (Gill and Niiler, 1973):

$$TS = \int_{-1500}^{0} \alpha T' dz \tag{1}$$

$$HS = -\int_{-1500}^{0} \beta S' dz \tag{2}$$

where α and β are the coefficients of thermal expansion and haline contraction, respectively, and the prime denotes deviations from the time-mean fields (i.e., anomalies). The integration is carried out over the vertical coordinate z. To obtain uncertainty estimates for TS and HS, we use Monte Carlo simulation to propagate uncertainties in T and S through Eqs. (1) and (2). This procedure involves first generating random profiles of T and S at each horizontal grid point and for each month under the assumption that the errors provided for the gridded data are normally distributed, where we allow for vertically correlated errors. Mathematically, the procedure would be expressed as follows:

$$T^{i} \sim N(\mu_{T}, \Sigma_{T}) \tag{3}$$

$$S^i \sim N(\mu_S, \Sigma_S) \tag{4}$$

where the superscript 'i' denotes a random profile (i = 1, ..., 100),  $\mu_T$  and  $\mu_S$  represent the means of the normal distributions which are set equal to the original gridded T and S values, and  $\Sigma_T$  and  $\Sigma_S$  are covariance matrices that encode the magnitude of the errors and their dependency in the vertical direction. The covariance matrices are specified by assuming an exponential model of the form  $\Sigma_{nm} = \sigma_n \sigma_m e^{-d_{nm}/l}$ , where  $d_{nm}$  is the distance between the n-th and m-th vertical levels,  $\sigma_n$  and  $\sigma_m$  are the error standard deviations provided by the gridded products at the corresponding levels, and l is a vertical decorrelation length scale. We set l equal to 100 m, based on the decorrelation length scales typically used in the objective analysis of Argo data (Good et al., 2013).



Then, for each random profile  $T^i$  and  $S^i$  we calculate the corresponding TS and HS values by evaluating Eqs. (1) and (2). This yields 100 random samples of TS and HS at each grid point and for each month. The standard deviations of the random samples provide an estimate of the uncertainties associated with the TS and HS fields.

The uncertainty estimates described above account only for uncertainty in the month-to-month variability.

However, we also want to account for uncertainty in the long-term trend. To quantify such uncertainty, we use T and S data from a second data set, the EN4 product (Good et al., 2013; available at <a href="https://www.metoffice.gov.uk/hadobs/en4">https://www.metoffice.gov.uk/hadobs/en4</a>), and derive trend uncertainty estimates from the spread across the ISAS20 and EN4 products (see Section 2.5 for more details).

Finally, we note that the contribution of waters below 1500 m to TS and HS will be inferred in the BHM by exploiting the relationship SL=TS+HS+OM and the fact that the SL from altimetry reflects the full-depth TS and HS contributions. To enable this, we inflate the uncertainty in the ISAS20-derived TS and HS by 20%, allowing for a deep-ocean (below 1500 m) TS and HS contribution.

## 2.2. SL from satellite altimetry




The altimetry sea-level data are from the gridded sea surface height product (based on a stable two-satellite constellation) (SEALEVEL\_GLO\_PHY\_CLIMATE\_L4\_MY\_008\_057) produced and distributed by the Copernicus Climate Change Service (C3S). These data are available at <a href="http://marine.copernicus.eu/">http://marine.copernicus.eu/</a> and are provided as daily fields on a 1/4° × 1/4° near global grid. For this study, C3S refers to the period from January 2004 to December 2020. The C3S data are provided with all standard corrections applied, including corrections for tropospheric (wet and dry) and ionospheric path delays, sea state bias, tides (solid earth, ocean, loading, and pole), and barotropic atmospheric effects (wind and atmospheric pressure for periods<20 days and inverse barometer effects for longer periods). We also adjust the sea-level fields for glacial isostatic adjustment (GIA) using the estimates derived by Frederikse et al. (2020) as well as for deformation effects on the sea floor (+0.1 mm yr<sup>-1</sup>, spatially uniform) due to contemporary mass changes of the Greenland and Antarctic ice sheets, glaciers, and terrestrial water storage (Frederikse et al., 2017).

The altimetry data are affected by several sources of uncertainty (Prandi et al., 2021), including mapping errors, high-frequency errors arising from both orbit determination and any of the geophysical corrections mentioned above, low-frequency errors associated with the wet tropospheric correction, drift errors from orbit determination, inter-mission biases (from Jason-1 to Jason-2 and from Jason-2 to Jason-3), and errors from the GIA adjustment. We account for all these error source contributions. The mapping errors are provided by C3S as part of the gridded product whereas estimates for the other error sources are from Prandi et al. (2021) (available at https://www.seanoe.org/data/00637/74862/). Details of how these uncertainties are taken into account in the data fusion analysis are given in Section 3.

## 2.3. OM from satellite gravimetry




The OM data are based on measurements collected by the Gravity Recovery and Climate Experiment (GRACE). Here, we use the global time-variable gravity mascon solution (RL06v2.0) from the NASA Goddard Space Flight Center (GSFC) (Loomis et al., 2019), available at https://earth.gsfc.nasa.gov/geo/data/grace-mascons. The data span the period from April 2003 to December 2021 and are provided in the form of monthly mascons on an equal-area 1° × 1° (at the equator) grid. Non-tidal ocean bottom pressure variations (GAD product) with their global ocean mean removed have been restored and a GIA correction has been applied, ensuring that the resulting OM data are comparable to the residual of altimetric SL and hydrography-derived TS+HS. The product supplies uncertainty estimates at each mascon, accounting for both serially uncorrelated errors due to leakage and stochastic noise and for leakage trends. In addition, we also account for errors in the OM long-term trend due to the GIA correction, geocentre motion and Earth oblateness (i.e., the degree 2 order 0 zonal spherical harmonic coefficient). Such errors are estimated based on the ensemble of global GRACE solutions from Blazquez et al. (2018), which is available at ftp.legos.obs-mip.fr/pub/soa/gravimetrie/grace legos.

## 2.4. Surface heat flux

The net surface HF is calculated by combining top-of-the-atmosphere (TOA) radiative flux with the divergence of vertical integral of total energy flux and the tendency of vertical integral of total energy as described in Mayer et al. (2022). The TOA flux has been obtained from the Clouds and the Earth's Radiant Energy System-Energy Balanced and Filled (CERES-EBAF) Edition-4.2.1 monthly data product (Loeb et al., 2018; available at 225 https://ceres.larc.nasa.gov/data), whereas the vertically integrated atmospheric energy quantities are those of Mayer and et al. (2022)have been obtained from the Copernicus Climate Data Store (https://doi.org/10.24381/cds.c2451f6b). Because neither of these datasets provides uncertainty estimates, we assess uncertainty in surface HF by comparing these estimates with HF derived from two reanalysis data sets: ERA5 (Hersbach et al., 2023; available at https://cds.climate.copernicus.eu) (Hersbach et al., 2023), and 230 NCEP/NCAR (Kalnay, 1996; available at https://psl.noaa.gov/data/reanalysis/reanalysis.shtml). The reanalysisbased HF across the air-sea interface is calculated as the sum of radiative and turbulent fluxes. While variability in HF is broadly consistent across the three data sets, differences in their time-mean HF values can be significant. Therefore, we estimate uncertainty separately for HF variability and the time-mean HF. Uncertainty in HF variability is calculated as the standard deviation (SD) of the HF anomalies (relative to the respective time means) 235 across the three products at each time step, whereas uncertainty in the time-mean HF is computed as the SD of the

long-term means from the three datasets. These SDs are calculated based on the spatially averaged HF time series over the budget regions (see Fig. 1). Note that the reanalysis-based HF data are used only for estimating uncertainty and are not directly incorporated into the BHM.

## 240 2.5. Resolving the spatial resolution mismatch





Here we aim to combine three data sets that are incompatible in terms of spatial resolution. Both the hydrographic data and the GRACE data are provided on relatively fine grids, but their effective spatial resolution is much lower as these data do not resolve features at the scale of the grid spacing. In particular, monthly GRACE data can be regarded as spatial averages over a ~300 km footprint (Tapley et al., 2019). The gridded hydrographic product is based on Argo profiles. Although such profiles represent point-level measurements, they are sparsely collected over the ocean with an average spatial separation of about 3 degrees (in any given month). This separation sets a practical limit to the smallest features that can be resolved (on average) by the Argo data. Additionally, the Argoderived gridded data incorporates some degree of spatial smoothing due to the interpolation process, which is largely determined by the decorrelation lengths scales used by the gridding methods. Such length scales are typically on the order of 300 km (Good et al., 2013), except within a few degrees of the equator where they are significantly larger. The effective resolution of the gridded altimetry data is higher than that of the GRACE and hydrographic data, although it varies with latitude. The decorrelation length scales used in the mapping of the altimetry data, which we take as a rough measure of spatial resolution, range from ~350 km in a low-latitude band (±15°N) to ~150 km poleward of this band (Ballarotta et al., 2019). In considering ways of resolving the resolution mismatch problem, it is important to remember that, here, we aim to assess heat budgets over large ocean regions and this only requires spatial averages of the relevant variables over such regions (i.e., we do not need point-level estimates). In this context, the simplest solution to the spatial support problem is to aggregate each of the three data sets into areal units (i.e., grid cells) of a size similar to the resolution of the coarsest resolution data. In this way, the aggregated data will all have the same level of spatial resolution and can then be combined using areal spatiotemporal modelling (Banerjee et al., 2014). In this study, we adopt this approach.

We set the size of the target areal units to  $3^{\circ} \times 3^{\circ}$  as this aligns with the decorrelation length scales used in the mapping of the hydrographic data as well as with the resolution of the GRACE data. For all three data sets, we aggregate the data by averaging the original grid cells over the target areal units through proportional area weighting (i.e., weights are computed as the proportion of the target areal unit that lies within each source grid cell).

The uncertainties in the aggregated data are computed by propagating the source data uncertainties through the spatial averaging process using standard error propagation formulae. In particular, the error variance of the aggregated data at any given target cell is calculated as:

$$\sigma^2 = \sum_{i=1}^n w_i^2 \sigma_i^2 + \sum_{i=1}^n \sum_{j \neq i}^n w_i w_j \rho_{ij} \sigma_i \sigma_j$$

$$\tag{5}$$

where  $w_i$  is the weight assigned to the *i*-th source grid cell in the aggregation process,  $\sigma_i$  is the standard error of the source data at the *i*-th grid cell, and  $\rho_{ij}$  is the error correlation between the *i*-th and *j*-th source grid cells. Note that the correlations  $\rho_{ij}$  are not known exactly and thus they need to be approximated somehow. Here, we assume a spatial correlation function of the form  $\rho_{ij} = e^{-d_{ij}/l}$ , where  $d_{ij}$  is the geodesic distance between the centroids of the *i*-th and *j*-th source grid cells, and *l* denotes a decorrelation length scale. For the GRACE data, we assume that errors from all sources are perfectly correlated in space. For the uncertainty in the TS and HS variability, we set l=300 km at latitudes poleward of 5° and l=600 km otherwise (based on the decorrelation length scales used in the mapping process). The uncertainty in the TS and HS trends (see Section 2.1) is calculated as  $|\beta_{ISAS20} - \beta_{EN4}|/\sqrt{2}$ , where  $\beta_{ISAS20}$  and  $\beta_{EN4}$  are, respectively, the trend estimates from ISAS20 and EN4 derived from the spatially averaged time series over each target areal units. Finally, for the altimetry data, we use different length scales depending on the type of error (see Section 2.2). For the interpolation errors, we set l=150 km at latitudes poleward of 15° and l=350 km otherwise (based on the length scales used in the original gridding of the data). The high-frequency errors are assumed to be spatially uncorrelated. For the errors and inter-mission biases are assumed to be perfectly correlated in space (over the length scales on which the spatial aggregation takes place).

## 2.6. Considerations on data temporal resolution






Although the data described above are available at monthly resolution, their effective temporal resolution is somewhat lower due to the relatively low sampling rate of the observations on which the gridded products are based. For example, most satellite altimeters have a repeat cycle that ranges from 10 days (e.g., Jason satellites) to 35 days (e.g., Envisat satellite), whereas GRACE has a repeat cycle of 30 days. Argo floats provide a vertical profile once every 10 days. In practice, this means that the month-to-month variability in the gridded products exhibits significant levels of noise. Such noise at high frequencies can be greatly amplified by the time derivatives involved in the calculation of MHT (the amplification factor is proportional to frequency) and, in turn, can corrupt our estimates of MHT. To minimize this issue, we trade off some temporal resolution for a significant reduction in noise by converting the monthly data to 3-month averages (i.e., to quarterly data: Jan-Feb-Mar, and so on).

The uncertainties associated with the quarterly data are calculated by propagating the uncertainties in the monthly data using Eq. (5), where now the sub-indices i and j refer to the i-th and j-th months, respectively. For the uncertainties in TS and HS variability as well as for the high-frequency errors in the GRACE and altimetry data (and the errors in the wet tropospheric correction) we set  $w_i = 1/3$  and assume serially uncorrelated errors ( $\rho_{ij} = 0$ ). In contrast, the leakage trends as well as the errors due to the GIA correction, geocentre motion and Earth oblateness in the GRACE data are, by definition, perfectly correlated in time and thus such errors are not reduced by the temporal averaging (their value remains the same). Similarly, for altimetry, the drift errors, inter-mission biases and GIA errors are all modelled as a linear trend error and thus they are also perfectly correlated in time. The uncertainties in the TS and HS trends are also perfectly correlated in time.

## 3. Bayesian hierarchical framework

Here, we develop a Bayesian hierarchical framework that integrates observations from altimetry, GRACE, and hydrography together with surface HF data to estimate HTC from ocean heat budgets over a set of latitudinally-bounded regions (see Fig.1). A schematic of the Bayesian model architecture is shown in Figure 2. The model is composed of two distinct but interconnected parts: one that models the spatiotemporal evolution of the sea-level components (TS, HS, and OM) based on the 3° × 3° areal units of quarterly (3-month-averaged) observations from hydrography, altimetry and GRACE; and another part that determines non-seasonal quarterly HTC as the residual of OHC tendency (derived using estimates of TS from the first component of the model) less surface HF over selected regions bounded by latitude lines (Fig. 1). The heat budgets are evaluated over all regions simultaneously, allowing for, but not enforcing, correlation in HTC between regions. It is important to note that the two parts described above are components of the same Bayesian model, i.e., we compute a single posterior distribution from which we derive expectation values (means and quantiles) for all the quantities of interest.

**Figure 2.** Schematic of the Bayesian hierarchical framework used to estimate MHT. The direction of the arrows in the Bayesian diagram reflects the flow of information in the generative process: from parameters/processes to observations.

Our approach centers on the notion that the data, being observed with error, can be regarded as the sum of a true latent spatiotemporal process plus observation error. For example, the GRACE data can be viewed as noisy observations of the OM latent process, and similarly for all the other data sets. Here, the term process refers to a stochastic process (i.e., a collection of random variables indexed by either time, space or both), whereas the term latent process means that the process is not directly measured but inferred from noisy observations. The aim is then to separate the various latent processes (TS, HS, OM, and HTC) from the observational noise through joint statistical modelling of all the available data (i.e., hydrography, altimetry, GRACE and surface HF). By conducting joint modelling and accounting for spatial dependencies, we allow for information exchange both between observational data sets and across locations, improving the estimation of the true underlying process. The success of this approach relies on the careful representation of uncertainty, not least because the model uses uncertainty to

decide which observations should have more influence. Furthermore, we need to consider not only uncertainty associated with measurement noise, but also uncertainty arising from limited knowledge of the latent process as well as uncertainty in unknown parameters of the BHM such as decorrelation length scales and error variances. This problem can be conveniently formulated within a Bayesian hierarchical framework.

Here, we develop a BHM with three layers (each layer takes the form of a probability model): 1) a data model that describes the distribution of the observations (SL, TS, HS, OM and HF) conditional on the latent processes; 2) a process model that describes the spatiotemporal evolution of the latent processes conditional on a set of parameters; and 3) a parameter model that describes the uncertainty in the model parameters and encodes any prior information that we have about the data and the processes. By modelling the latent processes probabilistically as random fields, we account for the fact that there are many plausible realizations of the underlying processes consistent with the observational data, thereby capturing uncertainty arising from limited knowledge of the latent processes. In our BHM, each component of sea level (i.e., TS, HS and OM) is modelled as the sum of three contributions, namely seasonal changes, variability, and a linear trend. The non-seasonal HTC is modelled as the combination of variability, a linear trend and an intercept. Next, we describe each layer of the BHM, starting with the data layer.

### 3.1. Observation layer.






Let  $B = \{B_i \mid i = 1, ..., N\}$  denote the set of  $3^\circ \times 3^\circ$  cells on which the observations have been aggregated, and  $z_{p,t}(B_i)$  denote an observation on the *i*-th cell at time step *t*, for  $p \in P$  where  $P = \{SL, TS, HS, OM\}$ . The data layer of the BHM for the sea-level observations can then be written as:

$$z_{p,t}(B_i) = y_{p,t}(B_i) + a_p(B_i) + b_p(B_i)t + e_{p,t}(B_i), \quad i = 1, ..., N \quad t = 1, ..., T$$
 (6)

where  $y_{p,t}(B_i)$  denote the true latent process of interest. The term  $a_p(B_i)$  represents data offsets, modelled as independent and identically distributed (i.i.d.) normal random variables with variance of 1 m²,  $a_p(B_i) \stackrel{iid}{\sim} N(0,1)$ . These offsets account for potential differences in vertical reference levels across the various observational data sets used in the analysis (hydrography, altimetry and GRACE). The variables  $b_p(B_i)$  denote data error trends, modelled as i.i.d. normal random variables,  $b_p(B_i) \stackrel{iid}{\sim} N(0, \gamma_p^2(B_i))$ . These include orbit errors, inter-mission biases and GIA uncertainties in the altimetry data, trend errors in the TS and HS data, and GIA, geocentre motion and Earth oblateness errors as well as leakage trends in the GRACE data. Finally,  $e_{p,t}(B_i)$  are assumed to be serially and spatially uncorrelated observation errors,  $e_{p,t}(B_i) \stackrel{ind}{\sim} N(0, \sigma_{p,t}^2(B_i))$ . The variances  $\gamma_p^2(B_i)$  and  $\sigma_{p,t}^2(B_i)$  are specified based on the data uncertainties calculated as described in Section 2.

Next, we define the data model for the surface HF data. Let  $R = \{R_j \mid j = 1, ..., M\}$  denote the set of regions over which the heat budgets are evaluated, and  $Q_t(R_j)$  be the non-seasonal surface HF into the ocean at time t spatially averaged over  $R_j$ . We express  $Q_t(R_j)$  as the sum of its time-mean value,  $\langle Q_t(R_j) \rangle$ , plus a fluctuating part,  $Q_t'(R_j)$ , as this allows us to model uncertainty in both components explicitly. With this, the observation layer for the HF data is written as a heat budget:

$$Q'_{t}(R_{j}) = H'_{t}(R_{j}) - U'_{t}(R_{j}) + v_{t}(R_{j}), \qquad j = 1, ..., M$$
(7)

$$\langle Q_t(R_i) \rangle = \langle H_t(R_i) - U_t(R_i) \rangle + q(R_i), \qquad j = 1, \dots, M$$
(8)

where  $H_t(R_j)$  is the non-seasonal OHC tendency spatially averaged over  $R_j$ ,  $U_t(R_j)$  is the HTC through the lateral boundaries of  $R_j$ , and  $v_t(R_j)$  and  $q(R_j)$  are serially and spatially uncorrelated observation errors,  $v_t(R_j) \stackrel{ind}{\sim} N(0, \eta_t^2(R_j))$  and  $q(R_j) \stackrel{ind}{\sim} N(0, \delta^2(R_j))$ . The SDs  $\eta_t(R_j)$  and  $\delta(R_j)$  are set equal to the SDs of the HF calculated as described in Section 2.4. The prime ' denotes deviation with respect to the time mean.

#### 3.2. Process layer.



Here, we describe the process layer of the BHM. Seasonal changes in the sea-level components are modelled as:

$$x_{p,t}^{\text{Seas}}(B_i) = a_{1,p}(B_i)\cos(\omega t) + a_{2,p}(B_i)\sin(\omega t) \qquad p \in P \setminus \{SL\}$$
(9)

where  $\omega$  is the angular velocity of the annual cycle, and  $a_{k,p}(B_i)$ , for k=1,2, are assumed to follow a spatial conditional autoregressive (CAR) process (Cressie and Wikle, 2011):

$$a_{k,p}(B_i) \sim MVN(0, (I_N - \alpha_{a,p}K)^{-1} \tau_{a,p}^2 D)$$
 (10)

where MVN denotes a multivariate normal distribution,  $I_N$  is the identity matrix of size N,  $\alpha_{a,p}$  is a parameter that controls the degree of spatial autocorrelation (to be estimated), K is the adjacency matrix ( $k_{ii} = 0$ ,  $k_{ij} = 1$  if  $B_i$  and  $B_j$  are neighbors, and  $k_{ij} = 0$  otherwise),  $\tau_{a,p}$  is a SD parameter (to be estimated), and  $D = diag(1/n_i)$  is an  $N \times N$  diagonal matrix with  $n_i$  equal to the number of neighbors of the i-th grid cell. Two cells are considered to be neighbors if the distance between their centroids is no larger than 7 degrees (i.e., roughly two times the size of the cells). CAR models are classes of Markov random fields commonly used to describe spatial autocorrelation in areally-aggregated data.

The non-seasonal variability in TS, HS and OM is modelled as spatial fields that evolve through time according to a first-order autoregressive moving average (ARMA) process. Here, we note that the TS and HS components of

sea level tend to covary inversely. To capture this spatiotemporal interaction between TS and HS, we use the method of coregionalization (Gelfand et al., 2004), which assumes that the interaction is local. With this, the model for the non-seasonal variability can be written as:

$$x_{p,t}^{\text{Var}}(B_i) = \rho_p x_{p,t-1}^{\text{Var}}(B_i) + \theta_p m_{p,t-1}(B_i) + m_{p,t}(B_i) - \psi m_{TS,t}(B_i) \mathbf{1}_{\{HS\}}(p) \qquad p \in P \setminus \{SL\}$$
 (11)

where  $\rho_p$  and  $\theta_p$ , for  $p \in P \setminus \{SL\}$ , are spatially constant coefficients that control, respectively, the degree of temporal autocorrelation and past-noise dependence,  $\psi$  is a positive parameter (to be estimated) that determines the strength of the interaction between the TS and HS fields, and  $m_{p,t}(B_i)$  are CAR processes,  $m_{p,t}(B_i) \sim MVN(0, (I_N - \alpha_{m,p}K)^{-1}\tau_{m,p}^2D)$ . The factor  $\mathbf{1}_A(x)$  is an indicator function such that  $\mathbf{1}_A(x) = 1$  if  $x \in A$ , and  $\mathbf{1}_A(x) = 0$  otherwise.

The linear trends in the latent sea-level processes are modelled as spatial fields, where again we capture the interaction between the TS and HS fields through coregionalization:

$$x_{p,t}^{\text{Trend}}(B_i) = \left(g_p(B_i) - \phi g_{TS}(B_i) \mathbf{1}_{\{HS\}}(p)\right)t, \qquad p \in P \setminus \{SL\}$$
(12)

where  $\phi$  is an interaction parameter (to be estimated), and  $g_p(B_i)$  are CAR processes,  $g_p(B_i) \sim MVN(\mu_{g,p}, (I_N - \alpha_{g,p}K)^{-1}\tau_{g,p}^2D)$ , for  $p \in P\setminus\{SL\}$ . The mean of the CAR process,  $\mu_{g,p}$ , is set to 0 for  $p \in \{TS, HS\}$  and to 2 mm yr<sup>-1</sup> for  $p \in \{OM\}$  based on the spatially averaged trends from the observational data (computed using ordinary least squares).

With all the contributions now defined, the true latent process for each sea-level component is given by:

$$y_{p,t}(B_i) = x_{p,t}^{\text{Seas}}(B_i) + x_{p,t}^{\text{Var}}(B_i) + x_{p,t}^{\text{Trend}}(B_i), \quad p \in P \setminus \{SL\}$$

$$\tag{13}$$

$$y_{SL,t}(B_i) = y_{TS,t}(B_i) + y_{HS,t}(B_i) + y_{OM,t}(B_i)$$
(14)

Note that  $y_{TS,t}$  and  $y_{HS,t}$  are the full-depth TS and HS contributions to total SL changes.

400

405

The non-seasonal OHC tendency spatially averaged over  $R_j$  is computed as follows. Let  $y_{TS,t}^{Des}(R_j)$  be the deseasonalized TS at time t spatially averaged over  $R_j$ , computed as:

$$y_{TS,t}^{Des}(R_j) = \frac{\sum_{i=1}^{N} w_{ij}(x_{TS,t}^{Var}(B_i) + x_{TS,t}^{Trend}(B_i))}{\sum_{i=1}^{N} w_{ij}}, \qquad j = 1, ..., M$$
(15)

where the weights  $w_{ij}$  are computed as the proportion of the region  $R_j$  area that lies within the grid cell  $B_i$ . The non-seasonal OHC tendency is then calculated using central differences as:

$$H_t(R_j) = \frac{\rho_0 c(R_j)}{\alpha(R_i)} \left( \frac{y_{TS,t+1}^{Des}(R_j) - y_{TS,t-1}^{Des}(R_j)}{2} \right), \qquad j = 1, \dots, M$$
 (16)

where  $\rho_0$  is a reference seawater density,  $c(R_j)$  is the time-mean heat capacity of seawater spatially averaged over  $R_j$ , and  $\alpha(R_j)$  is the time-mean coefficient of thermal expansion spatially averaged over  $R_j$  and vertically averaged over the top 1500 m of the ocean. Note that Eq. (16) assumes a constant  $\alpha$  to relate vertically integrated HTC to changes in TS. While this neglects vertical variations in  $\alpha$ , the approximation is necessary because modelling the relationship exactly would require knowledge of the vertical structure of HTC, which is not available. Furthermore, our Bayesian framework operates on two-dimensional, vertically integrated fields; incorporating vertical variations into the framework would significantly increase the model's complexity. Nevertheless, to assess the impact of this assumption, we tested the sensitivity of our estimates to different choices of  $\alpha$ , obtained by averaging over different depth ranges. The results show almost no effect on the phase of the estimated HTC variability and only a small effect on its amplitude. While this test does not fully rule out the possibility of small biases, it gives us confidence that the approximation is unlikely to introduce substantial errors.

Finally, the HTC through the lateral boundaries of  $R_j$ ,  $U_t(R_j)$ , is modelled as the sum of three contributions, namely variability, a linear trend, and an intercept. The variability in  $U_t(R_j)$  is allowed to be correlated across regions, and thus we modeled it as a spatial CAR process that evolves through time according to a first-order ARMA process:

$$u_t^{\text{Var}}(R_i) = \rho_U u_{t-1}^{\text{Var}}(R_i) + \theta_U f_{t-1}(R_i) + f_t(R_i)$$
(17)

where  $\rho_U$  and  $\theta_U$  are spatially constant coefficients that control, respectively, the degree of autocorrelation and past-noise dependence, and  $f_t(R_j)$  is a spatial CAR process  $f_t(R_j) \sim MVN(0, (I_M - \alpha_U L)^{-1} \tau_U^2 D_U)$ . Here, L is the  $M \times M$  adjacency matrix, and  $D_U = diag(1/n_j)$  is an  $M \times M$  diagonal matrix with  $n_j$  equal to the number of neighbors of the j-th region. In this case, two regions are considered to be neighbors if they share a common border. Both the linear trend and the intercept are modelled as i.i.d. normal random variables with SD of 1.3 W m<sup>-2</sup> yr<sup>-1</sup> and  $127 \text{ W m}^{-2}$ ,  $u^{\text{Trend}}(R_j)^{iid} \sim N(0, 1.3^2)$  and  $u^{\text{Intc}}(R_j)^{iid} \sim N(0, 127^2)$ .

With this, the HTC is calculated as:

$$U_t(R_j) = u_t^{\text{Var}}(R_j) + tu^{\text{Trend}}(R_j) + u^{\text{Intc}}(R_j).$$
(18)

## 3.3. Parameter layer


The BHM is completed by defining the parameter layer. The prior distributions ascribed to the hyperparameter of the BHM are summarized in Table 1. Here, we provide justification for the informative the priors. First, the priors assigned to the ARMA parameters ( $\rho_*$  and  $\theta_*$ ) ensure that the ARMA processes are stationary and invertible by enforcing the following conditions:  $0 < \rho_* < 1$  and  $|\theta_*| < 1$ . The constraint applied to the autocorrelation parameters of the CAR processes,  $\alpha_* < \lambda_{max}^{-1}$ , is necessary to ensure that the covariance matrix of the CAR processes is positive definite. The choice of the location parameter for the truncated normal distributions assigned to the SDs of the CAR trend processes (i.e.,  $\tau_{g,TS}$ ,  $\tau_{g,HS}$  and  $\tau_{g,OM}$ ) is based on the SD of the trend fields derived from the observations. Finally, the interaction parameters ( $\psi$  and  $\phi$ ) are assumed to be positive since the TS and HS components tend to be anticorrelated, with values likely ranging from 0 to 1.

**Table 1.** Prior distributions ascribed to the hyperparameters of the BHM. The constant  $\lambda_{max}$  is the maximum eigenvalue of the corresponding adjacency matrix (K for  $\alpha_{a,p}$ ,  $\alpha_{m,p}$  and  $\alpha_{g,p}$ , and L for  $\alpha_U$ ). The notation half- $\mathcal{N}$  denotes a half normal distribution whereas trunc- $\mathcal{N}(,)$ [a,b] represents a truncated normal distribution with support in the interval [a,b].

| Parameter                                                                 | Units               | Description            | Prior distribution                        |
|---------------------------------------------------------------------------|---------------------|------------------------|-------------------------------------------|
| $\rho_p \ p \in \{TS, HS, OM, U\}$                                        | -                   | ARMA autocorrelation   | Uniform(0,1)                              |
| $\theta_p \ p \in \{TS, HS, OM, U\}$                                      | -                   | ARMA past noise        | Uniform(-1,1)                             |
| $\alpha_{a,p},\alpha_{m,p},\alpha_{g,p},\alpha_{U}\ p\in \\ \{TS,HS,OM\}$ | -                   | CAR autocorrelation    | Uniform $(0,\lambda_{max}^{-1})$          |
| $\tau_{a,p}, \tau_{m,p}$ $p \in \{TS, HS, OM\}$                           | m                   | CAR standard deviation | half- $\mathcal{N}(0,1)$                  |
| $	au_{g,TS}$                                                              | mm yr <sup>-1</sup> | CAR standard deviation | trunc- $\mathcal{N}(3.5,1)[0,\infty]$     |
| $	au_{g,HS}$                                                              | mm yr <sup>-1</sup> | CAR standard deviation | trunc- $\mathcal{N}(2.0,0.3^2)[0,\infty]$ |
| $	au_{g,OM}$                                                              | mm yr <sup>-1</sup> | CAR standard deviation | trunc- $\mathcal{N}(1.3,0.3^2)[0,\infty]$ |
| $	au_U$                                                                   | W m <sup>-2</sup>   | CAR standard deviation | half- $\mathcal{N}(0,127^2)$              |
| $\psi$ , $\phi$                                                           | -                   | Interaction TS-HS      | half- $\mathcal{N}(0,1)$                  |

#### 3.4. Inference







Inference in the BHM is accomplished by numerically sampling from the posterior distribution of the processes and parameters given the observational data using the No-U-Turn Sampler (NUTS) as implemented by the Numpyro probabilistic programming language (Phan et al., 2019). NUTS is a Markov chain Monte Carlo (MCMC) method that uses Hamiltonian dynamics to enable rapid exploration of the posterior distribution. We run the sampler with four chains of 1000 iterations each (warm-up=1000) for a total of 4000 post-warm-up draws. Such draws represent samples from the posterior distribution.

#### 4. BHM evaluation

#### 4.1. MCMC diagnostics

We begin the evaluation of the model by assessing how accurately the MCMC samples characterize the posterior distribution. To this aim, we use a number of MCMC diagnostics that are designed to diagnose problems with the sampler and assess convergence and mixing. In this context, convergence means that the Markov chains have reached a stationary distribution that coincides with the true posterior distribution, whereas mixing refers to the number of iterations required for a Markov chain to approximate the posterior distribution adequately. While there are no definitive tests of convergence, we can use various diagnostics to determine whether Markov chains appear to have converged. One of such diagnostics is the potential scale reduction statistic (Gelman and Rubin, 1992),  $\hat{R}$ , which checks whether Markov chains initialized from different values have the same distribution (a necessary, although insufficient, condition for convergence). Mathematically,  $\hat{R}$  compares the sample variances both within and between Markov chains. When all the Markov chains have converged to the same distribution, values of  $\hat{R}$ should be close to 1 for all model parameters. While there is no universally accepted cut-off point for convergence based on  $\hat{R}$ , values of  $\hat{R} > 1.2$  are typically considered to be suggestive of non-convergence. In addition to convergence, evaluating the mixing of the Markov chains is also important as this can be poor in complex models due to high autocorrelation of the MCMC samples. The higher the autocorrelation the larger the MCMC standard error (given a fixed number of iterations), and thus the larger the error of the estimates derived from the posterior samples. As a measure of mixing and autocorrelation, we use estimates of the effective sample size,  $n_{\rm eff}$ , for each hyperparameter (Gelman et al., 2014). In general, a value of  $n_{\rm eff}$  per iteration <0.001 is indicative of poorly mixing chains and suggestive of possible biased estimates.

We find  $\hat{R}$  to be <1.06 for all hyperparameters in the BHM, suggesting that the Markov chains have converged to the equilibrium distribution and are providing a good approximation to the posterior distribution. Additionally, the

 $n_{\rm eff}$  per iteration is >0.01 for all hyperparameters with an average value of 0.43, indicating low autocorrelation and good mixing.

We use additional diagnostic tools, specific to Hamiltonian Monte Carlo, that offer information about the ability of the NUTS sampler to explore the posterior distribution. These tools include divergent transitions and tree-depth saturation. The presence of divergences would indicate that the sampler has run into regions of challenging posterior geometry that it is unable to explore well, whereas the appearance of tree-depth saturations would indicate that the sampler is terminating prematurely to avoid excessively long execution time thus leaving regions of the posterior distribution unexplored. We confirm that none of the iterations showed any divergent transitions or tree-depth saturations, giving us additional confidence in the reliability of our Bayesian solution.

## 4.2. Goodness of fit







Here, we evaluate the performance of our Bayesian model by examining its ability to accurately fit the observational data. We begin by examining the residuals from the observation model (Equation 6), focusing on the SL process as this integrates all the sea-level components (i.e.,  $z_{SL,t} - y_{SL,t} - a_{SL} - b_{SL}t$ ). The observation model assumes that the residuals are normally distributed, and thus gross violations of this assumption would signal the inadequacy of the model to describe the underlying structure of the data. It is, therefore, important to confirm the appropriateness of the normality assumption. Here, we do this by testing the null hypothesis that the residuals conform to a normal distribution using the Anderson-Darling test (Anderson & Darling, 1954). We apply the test separately to the time series of residuals at each grid cell and for each iteration. The test fails to reject the null hypothesis (5% significance level) in about 95% of the cases (4000 iterations x 641 cells), confirming the validity of the normality assumption. We also verify that there are no systematic departures between the Bayesian solution and the observations. In particular, we find that the time-mean of the residuals across all the grid cells are distributed symmetrically around zero, with a mean value of -0.1 mm and a SD of 3.4 mm.

Both the MCMC diagnostics and the residual analysis presented above indicate a good fit of the Bayesian model to the observational data. Despite this, it is still important to check that the posterior inferences from the model look plausible when compared to the observational data. We do this by plotting estimates of SL, TS, HS and OM against observational time series at a randomly selected grid cell (Fig. 3). All the sea-level components show considerable inter-annual variability, although such variability is significantly larger in the SL and the TS with peak-to-peak fluctuations that can reach more than 20 cm as compared to 8 cm for the HS and 2 cm for the OM. While the inter-annual variability is similar between the observations and the Bayesian solution, as we would expect, there are also significant differences between the two, especially for the TS and HS. These differences

reflect the relatively large uncertainties associated with the TS and HS observations and also demonstrate a crucial point: the ability of the BHM to constrain the TS and HS estimates based on information from the SL and OM data. This ability leads to more accurate estimates of TS and HS, in turn, allowing us to obtain more reliable estimates of MHT and other quantities of interest that depend on TS and HS.

For completeness, we have also plotted the linear trend estimates derived from the BHM on top of the time series (Fig. 3). In assessing the trends, it is important to note that the Bayesian trends are not directly comparable to least squares trends calculated from individual observed time series. Bayesian trends aim to capture the true underlying trend, free from the influence of ARMA variability and noise, whereas least squares trends will be affected by these factors. In practice, this means that the Bayesian trends will not necessarily be entirely aligned with what visually appears to be the tendency of the time series, although in general we would expect some degree of alignment, especially if the trend is large relative to the variability. In the example of the figure, the trends do agree to a large extent with the long-term tendency displayed by the time series, including in the OM time series for which the trend is the dominant signal.

**Figure 3.** Quarterly (3-month-averaged) time series of non-seasonal (a) sea level, (b) thermosteric, (c) halosteric and (d) ocean mass as derived from the BHM at a randomly selected grid cell (red) together with the corresponding

observed time series (black). Bayesian estimates of the linear trend at the same grid cell (blue) are also shown for the SL and its components. The shading around the red and blue lines represents the 5-95% credible interval.

#### 5. Atlantic meridional heat transport

#### 5.1. Calculation of the MHT






In this section, we describe how estimates of MHT over the Atlantic Ocean are derived from our Bayesian estimates of HTC. The BHM does not provide MHT directly as an output, but this can be calculated by meridionally integrating the HTCs. The integration can be started from any of the twelve latitude lines (see Fig. 1), but here we choose (for reasons that will become obvious later) to start from the northernmost latitude (i.e., 65°N). In this case, the northward heat transport across the i-th latitudinal section at time t, MHT $_{i,t}$  (using the same labeling as shown in Fig. 1), can be calculated as:

$$MHT_{i,t} = MHT_{1,t} + \sum_{i=1}^{i-1} HTC_{i,t}, \qquad i = 1, ..., M+1$$
(19)

where MHT<sub>1,t</sub> is the northward heat transport at 65°N and HTC<sub>j,t</sub> is the heat transport convergence in region  $R_j$  at time t (i.e.,  $U_t(R_j)$ ). Clearly, to evaluate Eq. (19) at any latitude line we need an estimate of MHT<sub>1,t</sub>, but such an estimate is not available. Let's now suppose that we set MHT<sub>1,t</sub> = 0 and then evaluate Eq. (19). This will produce estimates of MHT that will be accurate at any latitude where the true transport, MHT<sub>i,t</sub><sup>True</sup>, is large relative to the true transport at 65°N, i.e. where the following condition is met: MHT<sub>i,t</sub><sup>True</sup>  $\gg$  MHT<sub>i,t</sub><sup>True</sup>. While the validity of this approximation cannot be tested at all latitudes due to the lack of MHT estimates, it can be assessed at 26°N by comparing direct, independent estimates of MHT from the RAPID array (at 26°N) with those from the OSNAP array (at 50°N-60°N). Such comparison shows that the SD of the MHT time series (3-month means; overlapping period) is over four times larger at RAPID than at OSNAP, whereas the time-mean MHT is more comparable in magnitude with values of 1.19 PW (RAPID) and 0.51 PW (OSNAP). This indicates that ignoring the variability of MHT<sub>1</sub> in Eq. (19) will provide a very good approximation to the MHT variability at 26°N, assuming that the MHT variability at 65°N is similar to (or smaller than) that at the latitude of OSNAP. Ignoring the time-mean value of MHT<sub>1</sub> will, however, incur a larger error. Considering this, we approximate Eq. (19) as follows (we drop the temporal subscript t to simplify notation):

$$\mathsf{MHT}_i = \langle \mathsf{MHT}^{\mathsf{OSNAP}} \rangle - \langle \mathsf{HTC}_1 \rangle + \sum_{j=1}^{i-1} \mathsf{HTC}_j \;, \qquad i = 2, \dots, M+1 \tag{20}$$

where the angle brackets denote the time mean and  $\langle MHT^{OSNAP} \rangle$  is the time-mean MHT from the OSNAP array over the period 2014-2018 (i.e., it is set equal to 0.506 PW). Hence, in Eq. (20) we are essentially setting MHT<sub>1</sub> = 0 and the time-mean value of MHT at 60°N equal to  $\langle MHT^{OSNAP} \rangle$ . In making this approximation, we have implicitly assumed that the time-mean MHT at 60°N is constant over the period of our analysis, which appears to be a reasonable assumption based on the lack of a significant trend in the OSNAP MHT time series.

One might question why not to start integrating the HTCs at 26°N instead of at 65°N and then use the MHT from the RAPID array to set the integration constant, thereby avoiding any approximations. While this approach is indeed straightforward, it raises its own issues. First, our estimates would become directly dependent on the RAPID-derived MHT, leaving no independent observational estimates available to validate our results. Second, due to inherent uncertainties in both our calculations and the RAPID data, the meridionally integrated HTCs can never exactly match the RAPID MHT values. These discrepancies would propagate errors into the MHT estimates at all latitudes, with relatively larger errors occurring at latitudes where the MHT variability is small compared to that at 26°N, notably in the subpolar North Atlantic. Of course, the approximation used in Eq. (20) also introduces errors at these lower-variability latitudes, so it is important to determine which of these two approaches provides more reliable MHT estimates.

To assess this, we compare estimates of MHT in the North Atlantic derived using Eq. (20) with those derived from Eq. (21) and constrained using the RAPID-derived MHT (Fig. 4). To support the comparison, we also show the OSNAP-derived MHT time series at 60°N. At 26°N, the two Bayesian MHT estimates closely match each other in both magnitude and phase. However, the differences between them become more pronounced at higher latitudes. Specifically, the magnitude of the MHT variability in the Bayesian solution constrained by the RAPID-derived MHT remains nearly constant across latitudes. This contradicts direct observational evidence from the RAPID and OSNAP arrays, which shows that MHT variability at 26°N is more than four times greater than at 60°N. In contrast, the variability in the MHT derived from Eq. (20) decreases progressively towards higher latitudes and closely matches the magnitude of the MHT variability from OSNAP at 60°N, despite assuming zero variability at 65°N. These findings highlight two key points: (1) the approximation inherent in Eq. (20) holds to a good approximation; and (2) the solution obtained using Eq. (20) is more accurate than the one based on the RAPID-based constraint. In the following, we present the Bayesian MHT estimates derived using Eq. (20).

**Figure 4.** Quarterly (3-month-averaged) time series of MHT anomalies in the North Atlantic across the latitudinal sections denoted on the vertical axis as derived from the BHM by using Eq. (20) (red line) and by setting the MHT at 26°N equal to the RAPID-derived MHT (blue line). The OSNAP-derived MHT (yellow line) is also shown at 60°N.

#### 5.2. MHT at 26°N





Here, we compare our Bayesian estimates of MHT at 26°N calculated using Eq. (20) with the MHT from the RAPID array. In the following, any estimates derived from the BHM will be summarized by the posterior mean and the 5-95% credible interval (CI), where the CIs will be denoted by square brackets. The CIs are computed as the 5th to 95th percentiles of the posterior distribution, based on samples obtained from the BHM.

Before proceeding with the comparison, we note a recent study by Volkov et al. (2024) reporting on a spurious drift in the submarine cable measurements of the Florida Current that are used in the calculation of the RAPID-derived AMOC and MHT. In this study, they show that, after correcting for this spurious drift, the negative trend that is detectable in the uncorrected RAPID AMOC time series becomes barely statistically significant. This drift is also certain to affect the RAPID-derived MHT time series, but no correction for the MHT is publicly available at the time of writing this article. In view of this, we choose to remove a linear trend from the time series of MHT before the comparison to ensure that the comparison metrics reflect the true degree of concordance between the

Bayesian and the RAPID estimates. However, given the current debate on whether or not the AMOC is weakening, it is worth mentioning that our Bayesian estimate of MHT at 26°N shows no statistically significant trend for the period 2004-2020, with a value of 0.09 PW decade<sup>-1</sup> [-0.03,0.22]. Given the strong correlation between the AMOC and the MHT at 26°N (Johns et al., 2023), the lack of a statistically significant trend in the Bayesian solution appears to support the findings of Volkov et al. (2024).

Moving now to the comparison, we begin by showing the quarterly MHT time series as derived from the BHM together with the MHT from RAPID (Fig. 5a). Note that these time series represent 3-month-averaged values without any smoothing or scaling applied to them (i.e., they are the direct solutions from the BHM, except for the removal of a linear trend). Visually, it is already clear that the Bayesian solution accurately captures the variability in the RAPID-derived MHT, in terms of both the phase and the magnitude of the variability. This includes the prominent drop in MHT around 2010, the magnitude and timing of which are both captured with remarkable precision by the Bayesian solution. The most significant discrepancy between the two time series emerges after mid-2017, interestingly coinciding with the gap between the GRACE and GRACE Follow-On missions. More quantitatively, the correlation between the RAPID and Bayesian time series for the period from 2004 to early 2017 is high and statistically significant at the 95% confidence level with a value of 0.78. For the period 2004-2020, the correlation remains statistically significant but decreases slightly to 0.68. The SD of the MHT time series, which provides a measure of the magnitude of the variability, is also in excellent agreement between the Bayesian solution and the MHT from RAPID (Fig. 5b), with values of 0.18±0.03 PW for RAPID (90% frequentist confidence interval) and 0.17 PW [0.15,0.20] for the BHM.

**Figure 5.** (a) Quarterly (3-month-averaged) time series of MHT at 26° N derived using the BHM (red line) together with the RAPID-derived MHT (black line). The shading around the Bayesian time series represents the 5-95% CI. (b) SD of the MHT time series shown in panel (a), summarized by the mean (central mark), interquartile range (box) and 5–95% CI (whiskers). (c) Time-mean of the MHT time series shown in panel (a), with the boxes and whiskers defined as in (b). The quantiles for the BHM solution have been computed from the posterior distribution, whereas for the RAPID-derived time series they have been calculated based on the uncertainty provided by Johns et al. (2023), assuming a normal distribution. The CIs associated with the Bayesian estimates of the SD and the time-mean MHT reflect both statistical uncertainty due to the variability of the MHT time series as well as uncertainty from all sources considered in the BHM.

620

625

Focusing now on the time-mean MHT, we can already sense from the comparison of the time series (Fig. 5a) that the Byesian solution matches the time-mean MHT from RAPID very well. This visual intuition is confirmed by calculating the time-mean value of the time series and its uncertainty for the period 2004-2020 (Fig. 5c). The time-mean MHT from RAPID is 1.19±0.20 PW (90% confidence interval), whereas for the BHM we obtain a value of 1.14 PW [1.01,1.27], thus confirming an almost perfect agreement.

Next, we evaluate the agreement between the Bayesian solution and the RAPID-derived MHT at lower frequencies by applying a 4-quarter running mean to the MHT time series (Fig. 6). This is important since agreement at the

quarterly scale does not necessarily imply agreement at lower frequencies. At these lower frequencies, the agreement with RAPID is also remarkably good, with a correlation of 0.92 for the period 2004 to early 2017. For the period of 2004-2020, the correlation is slightly lower but remains high with a value of 0.81. The BHM-derived MHT time series closely tracks both the timing and magnitude of many prominent features in the RAPID data, including the sharp drop around 2010 and the subsequent recovery. The main discrepancy arises after 2017, when the Bayesian estimate shows a gradual decline in MHT whereas the RAPID time series displays a rapid increase during 2017-2018, followed by an even steeper decline in 2019. A smaller mismatch also occurs during 2005–2007, when the RAPID-derived MHT exhibits higher MHT values compared to the Bayesian estiate, but such differences fall within the Bayesian uncertainty band. Furthermore, the elevated MHT values in the RAPID time series in this period are expected to decrease following the application of the drift correction proposed by Volkov et al. (2024), which would likely further improve the agreement between the RAPID time series and the Bayesian estimate.

**Figure 6.** Low-pass filtered (4-quarter running mean) time series of MHT at 26° N derived using the BHM (red line) together with the RAPID-derived MHT (black line). The shading around the Bayesian time series represents the 5-95% CI.

Earlier in this paper, we argued that deriving MHT from hydrographic observations alone is likely to introduce significant errors into the MHT estimates due to the sparseness of these observations, and claimed that these difficulties can be overcome by incorporating SL and OM data from satellites. To support this claim, we have derived the MHT at 26°N from Eq. (20) using only hydrographic data (the same data as used in the BHM) and compared it with the RAPID-derived MHT (Fig. A1). It is immediately clear that the agreement with the MHT from RAPID is significantly worse for the hydrography-only based estimates than for the BHM estimates, with correlations of 0.39 and 0.55 for the quarterly and low-pass filtered time series, respectively. Furthermore, the

variability in the hydrography-only based estimates is significantly larger than in both the RAPID and BHM estimates, reflecting errors arising from the sparseness of the hydrographic data.

#### 5.3. MHT over the Atlantic Ocean

Although the excellent agreement of the Bayesian estimate with the MHT from RAPID at 26°N does not guarantee a similar performance at other latitudes, it does give us confidence that the estimates are robust and likely to reflect true changes. In this section we show our estimates of MHT across eleven latitude lines over the Atlantic (we do not include 65°N as our estimates are set to zero at this latitude), both in their quarterly (3-month-averaged) resolution (Fig. 7a) and after applying a 4-quarter running mean (Fig. 7b). Note that, unlike for the comparison with the MHT from RAPID, here we do not remove a linear trend from any of the time series.

Looking at the quarterly MHT time series (Fig. 7a), we observe a number of distinct features. The amplitude of the MHT fluctuations tends to increase gradually from north to south. For example, taking the SD of the time series as a measure of variability, we find that SD values range from 0.02 PW [0.02,0.03] at 60° N to a maximum value of 0.31 PW [0.26,0.35] at 35° S. Peak-to-peak fluctuations can be as large as 0.8 PW at 26° N and 1.5 PW at both 35° S and 25° S. The MHT fluctuations show a strong latitudinal coherence within two bands of latitudes, namely between 35° N and 16° N and from 5° N to 35° S. Such bands are not entirely decoupled from one another, but the MHT coherence within a band is much stronger than between bands. In particular, the event of 2010, which is so prominent in the MHT time series at 26° N, is also clearly visible at 16° N but much less so at 5° N. The lack of coherence south of 16°N during this event can be attributed to its main underlying cause – a southward shift of the North Atlantic subtropical gyre driven by a southward displacement of the mid-latitude westerlies (Evans et al., 2017) – the effects of which were largely confined to the mid-latitude regions with limited influence further south. Turning now our attention to the low-pass filtered time series (Fig. 7b), we note again that the magnitude of the fluctuations increases southward, although differences in magnitude across latitudes are smaller. The bands of latitudinal coherence are essentially the same as noted for the quarterly values.

**Figure 7.** (a) Quarterly (3-month-averaged) time series of MHT anomalies across the latitudinal sections denoted on the vertical axis as estimated using the BHM. (b) Same as panel (a) but with a 5-quarter running mean applied to the time series.

Finally, it is also interesting to plot the time-mean MHT from the BHM across all the latitudinal sections of the Atlantic (Fig. 8). For comparison, observation-based estimates from the RAPID array (at 26° N), Trenberth et al. (2019) as denoted by T19, and Liu et al. (2022) as denoted by Liu22 are also shown. The time-mean MHT is computed over the period 2004-2020 for the Bayesian solution and RAPID, 2004-2016 for T19, and 2006-2013 for Liu22. Focusing first on the Bayesian solution, we note that the time-mean MHT is positive, and thus northward, at all latitudes, achieving its maximum value around 26° N. The time-mean MHT has a value of 0.51 PW [0.49,0.52]

at 60° N (recall this value is set to match the time-mean MHT from the OSNAP array), which gradually increases southward reaching a maximum value of 1.14 PW [1.01,1.27] at 26° N, before decreasing again to a minimum value of 0.31 PW [0.06,0.54] at 35° S. Note that the Bayesian CIs tend to become wider as we move southwards. This reflects both increased MHT variability as well as larger differences between surface HF products (recall that the uncertainty in the surface HF data has been calculated as the spread over the three HF products). The latitudinal structure of the time-mean MHT from the Bayesian solution closely resembles that of other observation-based estimates (RAPID, T19 and Liu22). In the North Atlantic, the Bayesian estimates generally fall between those of T19 and Liu22. South of the equator, the Bayesian MHT is lower than both T19 and Liu22, though it remains very close to T19 and consistent with Liu22 when accounting for uncertainty ranges. The differences between the BHM solution and other estimates may arise from the fact that they cover slightly different time periods. To evaluate this, we calculated the time-mean MHT from the BHM over the Liu22 period (2006-2013) and compared it to BHM estimates for the period 2004-2020. The differences in the time-mean MHT between the two periods are less than 10% of the time-mean MHT at all latitudes, suggesting that the discrepancies with T19 and Liu22 are unlikely to be due to differences in averaging periods alone. Methodological and data-related differences are therefore a more likely explanation.

**Figure 8.** Time-mean MHT derived from the BHM along with the associated 5-95% CIs (whiskers) at multiple latitudes of the Atlantic Ocean. The CIs reflect both statistical uncertainty due to the variability of the MHT time series as well as uncertainty from all sources considered in the BHM. Observation-based estimates from the RAPID array, Trenberth et al. (2019) as denoted by T19, and Liu et al. (2022) as denoted by Liu22 are also shown. The confidence interval associated with the RAPID (90% interval) has been derived from the values provided by Johns et al. (2023).

## 7. Conclusions

Here, we have generated observation-based probabilistic estimates of MHT for the period 2004-2020 at 3-month-mean resolution across 12 latitudinal sections of the Atlantic Ocean between 65° N and 35° S. The MHT has been calculated based on estimates of HTC which, in turn, have been inferred as a residual from the difference between OHC tendency and surface HF through joint spatiotemporal modelling of observations from hydrography, satellite altimetry and satellite gravimetry.

Our estimates of MHT agree remarkably well with estimates based on direct ocean observations from the RAPID array at 26° N, capturing both the magnitude and phase of the MHT variability in the RAPID time series with high

fidelity. Compared with previous studies, our approach yields higher correlations with the RAPID-derived MHT. For example, Meyssignac et al. (2024) reported a correlation of 0.54 for the period 2005-2018, while we find a correlation of 0.85 for the same period when applying a 4-quarter low-pass filter. Similarly, Liu et al. (2020) found a correlation of 0.66 for 2008-2016, compared to 0.93 for the BHM-derived MHT. For 2004-2016, Mayers et al. (2022) reported a correlation of 0.72, whereas our estimate reaches 0.92 for that period. Note that all correlations reported in these previous studies are based on annual or 12-month low-pass filtered data, consistent with the temporal smoothing we apply in our comparisons. Part of the reason for the higher correlations may be because we focus on detrended time series to remove the influence of a known drift in the RAPID-derived MHT, whereas previous studies may have included linear trends in their correlation estimates. However, the improvements also likely reflect methodological differences, particularly our use of a BHM to account for spatiotemporal dependencies when combining hydrographic and satellite measurements.

We have also found that the magnitude of the variability tends to grow gradually from north to south, with peak-to-peak fluctuations that can reach 1.5 PW at the southernmost latitudinal sections. The MHT variability is not coherent across the whole Atlantic Ocean but instead covaries within, although not between, two distinct bands of latitudes, namely between 35° N and 16° N and from 5° N to 35° S. Regarding the time-mean MHT, our Bayesian solution shows northward MHT across the entire Atlantic Ocean, with a latitudinal profile that is characterized by relatively small transport in the South Atlantic Ocean, particularly south of 11° S, and maximum transport around 26° N.

Our results are exceptionally good if we accept the agreement with RAPID-derived MHT as a measure of performance, but there are some limitations that the users of these estimates need to be aware of. The first one is that our estimates of MHT have been derived by assuming no variability in the MHT at 65° N. As discussed earlier, this assumption will produce very accurate estimates of MHT variability over most of the Atlantic Ocean, but the incurred errors will be larger at high latitudes of the North Atlantic where the MHT variability is smaller in magnitude, particularly north of 45° N. Additionally, while estimates of the time-mean MHT are not affected by this assumption, it is important to remember that the time-mean MHT at 60° N has been set equal to that derived from the OSNAP array over the period 2014-2018. This assumes a constant time-mean MHT at 60° N over the analysis period (2004-2020), and thus deviations from this premise will introduce an error into the time-mean MHT at all the other latitudes. That said, such an error is expected to be very small based on the absence of a trend in the MHT time series from OSNAP and the excellent agreement with the RAPID-derived MHT at 26° N. Note also that our Bayesian model operates in two dimensions and therefore uses a vertically integrated thermal expansion coefficient to relate changes in TS to the vertically integrated HTC, which may introduce some approximation

error. Finally, it is also important to note that our estimates are derived from observational data spatially aggregated into  $3^{\circ} \times 3^{\circ}$  areal units, hence our approach is unlikely to resolve variations in MHT between latitude lines spaced by less than  $3^{\circ}$ .

# **Appendix A: Additional figures**

750

760

765

**Figure A1**. (a) Quarterly (3-month-averaged) time series of MHT at 26° N derived from Eq. (20) using only hydrographic data (red line) together with the RAPID-derived MHT (black line). The shading around the Bayesian time series represents the 5-95% CI. (b) The same as (a) but with a 4-quarter running mean applied.

# Data availability

The Bayesian estimates of MHT and HTC are available at Zenodo via https://doi.org/10.5281/zenodo.16640426.

**Author contribution:** FMC and EFW conceived and designed the study. FMC developed and implemented the Bayesian hierarchical model, with input from EFW and PV. FMC performed the analyses and wrote the manuscript, with contributions from EFW and PV. All authors participated in the interpretation of the results and commented on the submitted manuscript.

**Acknowledgements:** We thank the researchers and projects who made the data used in this study publicly available, including A. Blazquez for providing the GRACE uncertainty estimates and C. Liu for sharing their estimates of MHT. FMC, EFW and PV were supported by the EPOC project funded by the European Union's Horizon Europe programme (grant agreement No 101059547), under call HORIZON-CL6-2021-CLIMATE01.

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
