# Peer review of "Estimates of Atlantic meridional heat transport from spatiotemporal fusion of Argo, altimetry and gravimetry data"

_EGUsphere, 2025_

## Author Comment (AC1)

**Response letter**

We thank the reviewers for their valuable and constructive comments. The reviewers have raised important points that provide us with an opportunity to clarify a number of aspects relating to our results and the Bayesian model. In the following, we explain how the manuscript has been changed and provide a point-by-point response to each of the reviewers' comments. For clarity, our responses are in standard font whereas any text from the reviews is denoted by blue font. Text from the manuscript is italicized.

The main changes made to the manuscript are outlined below:

1. We use a new observation-based surface heat flux data set and focus our analysis on the Bayesian estimates derived from this new data set, in response to a comment by Reviewer #2. Accordingly, we no longer include Bayesian estimates based on ERA5 and NCEP data.
2. We now account for errors in the ocean mass from GRACE due to the GIA correction, geocentre motion and Earth oblateness.
3. We provide a clearer justification for our assumption of zero MHT at 65ºN, in response to a comment by Reviewer #1.

**Referee #1:**

This paper aims to estimate the meridional heat transport (MHT) at transatlantic sections throughout the Atlantic Ocean. Mainly, it uses hydrographic and satellite data via a Bayesian hierarchical model (BHM) to calculate the ocean heat content (OHC) tendency. The latter is then combined with air-sea heat flux data product to derive the ocean heat divergence and the MHT (as a residual from heat budgets). Accurate MHT estimates are critical for understanding the ocean's role in our climate system. Overall, the paper reads well, and the results are presented. However, there is potential confusion about the goals and motivations of this study, which would make it hard to follow what is presented and what one can learn from it. I recommend it for publication after the following minor comments are addressed.

We thank the reviewer for their positive assessment of our manuscript and their comments and suggestions.

Main comments:

The main goal of this paper is to provide MHT estimates that maximize the use of hydrographic and satellite data, via a new framework (BHM). However, I have difficulties in understanding the argument of not using the RAPID data to derive the MHT at other latitudes. Instead, the authors make assumptions about the MHT at the northern boundaries, which introduce uncertainties in the MHT estimates across all latitudes. In my opinion, it undermines the deliverables from this study.

We agree with the reviewer that the argument for not using the MHT from RAPID to derive the MHT at other latitudes could have been made more convincingly in the original manuscript. We would like to assure the reviewer that we considered this issue carefully during the development and testing of the Bayesian model. Based on numerous tests, we concluded that assuming no variability in MHT at 65º N leads to more reliable estimates at high latitudes.

It is important to note that setting the MHT at 26º N equal to the RAPID-derived MHT can also introduce errors at all latitudes, as we discussed in the manuscript. These errors arise for two main reasons. First, any biases in the RAPID-derived MHT, such as the spurious drift reported by Volkov et al. (2024), will propagate throughout the Bayesian estimates at all latitudes. Second, inconsistencies between the RAPID MHT and the Bayesian solutions can also introduce significant errors, especially at higher latitudes where the MHT variability is smaller. Even small mismatches at 26°N can appear as large errors at higher latitudes. This is precisely what our tests demonstrated.

In one such test, we compared the two Bayesian estimates of MHT at several latitudes in the North

Atlantic, and also compared both estimates with the OSNAP-derived MHT at 60° N. Following the reviewer's comment, we have added the results of this comparison to the revised manuscript, including a new Figure (Fig. 4) and the following paragraph:

*"At 26°N, the two Bayesian MHT estimates closely match each other in both magnitude and phase. However, the differences between them become more pronounced at higher latitudes. Specifically, the magnitude of the MHT variability in the Bayesian solution constrained by the RAPID-derived MHT remains nearly constant across latitudes. This contradicts direct observational evidence from the RAPID and OSNAP arrays, which shows that MHT variability at 26°N is more than four times greater than at 60°N. In contrast, the variability in the MHT derived from Eq. (20) decreases progressively towards higher latitudes and closely matches the magnitude of the MHT variability from OSNAP at 60°N, despite assuming zero variability at 65°N. These findings highlight two key points: (1) the approximation inherent in Eq. (20) holds to a good approximation; and (2) the solution obtained using Eq. (20) is more accurate than the one based on the RAPID-based constraint."*

These results together with the good agreement of our estimates with the RAPID-derived MHT at 26°N give us confidence in the robustness of our estimates.

Another source of uncertainty in the MHT estimates is from surface heat flux. However, surface heat flux itself likely contains larger uncertainty than the heat divergence derived from this study – as that is indicated by the discrepancies between BHM solutions. I would suggest that the authors provide a thorough uncertainty estimate that takes into consideration errors in surface heat flux.

We would like to clarify that observation errors are already accounted for in all data sources, including the surface heat flux (HF) data. It is not entirely clear to us what additional steps the reviewer is suggesting beyond what is already implemented. As noted in the manuscript, none of the HF products provides uncertainty estimates, thus we derive a measure of uncertainty from the spread across three HF products. See Section 2.4 for more details. These uncertainties, along with those from other data sources, are incorporated into the BHM, which enables a comprehensive treatment of uncertainty with rigorous error propagation.

If the goal of this paper is to prove the efficacy of the new BHM framework that combines hydrographic and satellite observations, should it be compared with one that just uses hydrographic data? That would highlight the advantages of the BHM.

We agree with the reviewer's suggestion and have responded by including a comparison between the RAPID-derived MHT at 26° N and the MHT derived using the budget approach based only on the hydrographic data. The results are discussed in Section 5.2 and include a new figure (Fig. A1).

This is related to comment#3. Much of this paper is centered on the discrepancies between BHM solutions (see Figures 4, 5, 6, 7 and the related text). Those comparisons are useful as an evaluation of how different surface heat flux data impact the MHT estimates. But such an evaluation itself is not well motivated. In addition, the MHT estimates are validated against Trenberth et al. (2019). But it is not clear we gain from this analysis that is distinct from Trenberth et al. which uses atmospheric reanalyses (surface heat flux) and hydrographic data (the OHC tendency).

As mentioned earlier in this response letter, the revised manuscript includes only estimates based on a new surface HF product. Following a comment by Reviewer #2, the comparison with Trenberth et al. (2019) is now limited to the time-mean MHT.

To answer the reviewer's question, our estimates differ from those of Trenberth et al. (2019) in that they are based on hydrographic observations constrained by satellite data in a statistically rigorous manner, rather than on ocean temperature fields from reanalyses. Additionally, we use a more recent surface heat flux product and provide quarterly estimates, whereas Trenberth et al. (2019) report 12-month filtered values.

Line 90: TS and HS are anomalies relative to the climatology density. Other terms should also be anomalies? Please be specific about each term.

Done.

Line 133: 'interesting oceanographically' reads odd.

This sentence has been reworded.

Line 148: How large is the volume transport? If it is large, it affects the mass conservation and thus the MHT estimate. Such effects on the related sections need to be discussed.

Originally, we did not exclude the Gulf of Mexico (GoM) and the Caribbean Sea from our analysis. However, during testing of the BHM, we identified issues with the ISAS20 data in these regions. For example, as shown in Fig. R1, the thermosteric sea level from a grid cell in the Caribbean Sea displays anomalous variability prior to 2011, suggesting potential problems in the data. While the BHM may partially correct for such issues through constraints from satellite observations, we ultimately chose to exclude these regions to avoid introducing spurious signals.

[Figure]

**Figure R1**. Thermosteric sea level from one of the grid cells in the Caribbean Sea.

In response to the reviewer's comment, we have assessed the impact of this exclusion by comparing MHT estimates from two BHM solutions: one that includes the GoM and Caribbean Sea, and one that excludes them as in the paper (see Figs. R2 and R3). The differences between the two are minimal, both in terms of MHT variability (Fig. R2) and time-mean MHT (Fig. R3), indicating that excluding these regions has a negligible effect on our results.

We have added the following sentence to the revised manuscript to clarify this point:

*"We have tested the impact of excluding data in these regions from on our estimates of MHT and found it to be minimal."*

[Figure]

**Figure R2**. Quarterly (3-month-averaged) time series of MHT anomalies across the latitudinal sections denoted on the vertical axis as estimated using the BHM for two cases: 1) excluding data from the Gulf of Mexico and the Caribbean Sea (red line); and 2) including data in these regions (blue line).

[Figure]

**Figure R3**. Time-mean MHT along with the associated 5-95% CIs (whiskers) at multiple latitudes of the Atlantic Ocean derived from the BHM for two cases: 1) excluding data from the Gulf of Mexico and the Caribbean Sea (red line); and 2) including data in these regions (blue line).

Line 172: Uniform l= 100m spatially and vertically? How valid are such assumptions?

The decorrelation length scale for errors in the gridded temperature and salinity (T and S) fields is not provided in the ISAS20 product, which only includes error variances but not covariances. While the vertical error correlation structure is unknown, some degree of vertical dependence is expected, as the T and S data are derived from vertical profiles. We therefore consider it reasonable to set the vertical decorrelation length scale equal to the length scales used in the objective analysis of the profiles themselves. We believe that this choice provides a sensible estimate in the absence of explicit error correlation information.

**Line 215: Are the two reanalysis products only used onward 12/2017? If yes, how?**

Please note that in the revised version of the manuscript, we compute a single Bayesian solution based on a new observation-based surface heat flux data set. The ERA5 and NCEP heat flux products are used only to obtain a measure of uncertainty in the surface heat flux data; they are not used directly in the BHM. This has been clarified in the revised manuscript.

**Line 218: Are the reanalysis products averaged together with DEEP-C? This appears to contradict the previous statement that 'it is preferable to' the reanalysis products.**

Please see our response to the previous comment. But to answer the reviewer's question, no, the reanalysis products were not averaged together with DEEP-C in the original submission. Instead, they were used to characterize the uncertainty in the surface heat flux data.

**Line 228: 'effective spatial resolution is much lower than what such grids imply' Hard to understand what it means – please reword.**

We have reworded the sentence for clarity. What we meant is that although some products are provided on relatively fine spatial grids, this does not mean they can reliably resolve features at those grid scales. The actual effective spatial resolution (i.e., the smallest scale at which meaningful information is retained) is often coarser than the grid spacing suggests. The revised sentence reads:

> *"Both the hydrographic data and the GRACE data are provided on relatively fine grids, but their effective spatial resolution is much lower as these data do not resolve features at the scale of the grid spacing."*

**Line 240: If the goal is for an integrated value over the region between two latitudes (11 regions in total, Fig. 1), why does one need spatial grids anyway? Why not consider the enclosed basin as a whole?**

We apply the BHM at the spatial grid level because this allows us to leverage cross-variable spatial information (across SL, TS, HS and OM) as well as to account for spatial error structures. This leads to significantly improved estimates of thermosteric sea level at each grid cell. These improved gridded fields can then be used to produce more accurate spatial averages over each budget region.

Our concern with computing regional averages directly from the original, unconstrained thermosteric fields is that such averages are likely to be biased due to the sparseness of the temperature data. Once the data are spatially averaged, it becomes very difficult to correct for these biases as it is no longer possible to exploit spatial information. Working in the gridded domain allows us to apply a coherent spatiotemporal modelling framework that reduces such biases and provides a more principled approach to uncertainty estimation.

We explain this in the manuscript, for example:

> *"By first calculating spatial averages separately for each variable, the procedure ignores any spatial dependencies between the variables and loses the opportunity to leverage cross-variable spatial information, both of which can lead to suboptimal estimates of spatially averaged values. Also, such a modelling choice makes the estimation of uncertainties in the spatially averaged values challenging, often requiring ad-hoc or approximate methods."*

> *"Our approach extends that of Kelly et al. (2016) by accounting for spatiotemporal dependencies between processes (i.e., TS, HS, and OM) and enabling information sharing*

*across the various data sets. This is achieved by simultaneous spatiotemporal modelling of the observational fields and their error structures, in contrast to time series modelling of spatially averaged values as done in Kelly et al. (2016)."*

Line 281: Setting rhoij= 0 requires justifications. The decorrelation time scale should be evaluated separately for each dataset, which is likely longer than a month.

The decorrelation time scale referred to by the reviewer is only used in our analysis to propagate observational uncertainties from monthly to quarterly values. In response to the reviewer's comment, we have assessed the sensitivity of our Bayesian MHT estimates to different values of $\rho_{ij}$.

The most conservative assumption, perfect temporal correlation, corresponds to $\rho_{ij}=1$, in which case the standard errors on the quarterly values are $\sqrt{3}$ times larger than in the uncorrelated case ($\rho_{ij}=0$). We have compared the resulting MHT estimates for both cases (see Fig. R4). The two estimates are nearly indistinguishable, with the main difference being a modest increase in the uncertainty of the MHT estimates for $\rho_{ij}=1$, by a factor of 1.07 on average. This limited sensitivity is expected, as changing $\rho_{ij}$ affects only the magnitude of the observation errors, not their spatial structure, which is the dominant sources of constraint in the BHM.

Based on this assessment, we have decided to retain our original choice of $\rho_{ij}=0$, which simplifies the calculation without significantly impacting the results.

[Figure]

**Figure R4**. Quarterly (3-month-averaged) time series of MHT anomalies across the latitudinal sections denoted on the vertical axis as estimated using the BHM for two cases: 1) $\rho_{ij}=0$ (red line); and 2) $\rho_{ij}=1$ (blue line).

Line 288 and Figure 2: What are the different arrows in Figure 2? For example, those black arrows

within the right red box indicate that Q is derived by H minus HTC. But that is opposite to what's described in the text.

We thank the reviewer for this question. In Bayesian diagrams, the arrows represent how we believe the observations were generated. That is, we assume there is an underlying latent process (e.g., the true thermosteric field), and the observations (e.g., instrumental readings) are noisy or indirect measurements of that process. The direction of the arrows reflects the flow of information in the generative process: from parameters/processes to observations. This convention also emphasizes the idea that uncertainty in parameters/processes propagates to uncertainty in the observations. Once we have observed the data, we then use Bayesian inference to reason backward through the arrows to estimate the latent processes.

In response to this comment, we have added the following sentence to the caption of Fig. 2:

> *"The direction of the arrows in the Bayesian diagram reflects the flow of information in the generative process: from parameters/processes to observations."*

Line 298: Why are the reanalysis products used separately? This is related to my comment above.

Please note that this is no longer the case in the revised manuscript. See comment at the beginning of this response letter and also our response to an earlier comment.

Line 315: How exactly are uncertainties determined? It is the key to providing a meaningful estimate.

We thank the reviewer for this important question. In a BHM, uncertainties are explicitly modelled and quantified through the probabilistic structure of the model. Uncertainty in the underlying latent processes is captured by modelling them as stochastic processes. By modelling the latent processes probabilistically as spatiotemporal random fields, we quantify the fact that there are many plausible versions of the underlying fields consistent with the data and prior knowledge. This is what we refer to as uncertainty arising from limited knowledge of the latent process. Uncertainty in model parameters (such as decorrelation length scales) is captured by assigning prior distributions to these parameters. Observation uncertainty is incorporated through the data likelihood, by specifying a distribution for the measurement error. The Bayesian framework allows us to coherently propagate all these sources of uncertainty into the posterior distributions of the quantities of interest.

In response to the reviewer's comment, we have added the following sentence to the revised manuscript:

> *"By modelling the latent processes probabilistically as random fields, we account for the fact that there are many plausible realizations of the underlying processes consistent with the observational data, thereby capturing uncertainty arising from limited knowledge of the latent processes."*

Line 520: What does it mean by 'will be accurate at any latitude'? How to quantify this accuracy? Also, why is the true transport at a given latitude is 'large relative to the true transport at 65N'?

Thank you for the opportunity to clarify this point. As shown in Eq. (20), the MHT at any latitude is computed as the sum of the MHT at 65ºN and the cumulative integral of heat transport convergences (HTCs) from 65ºN to the latitude of interest. Therefore, if the second term (i.e., the integrated HTC) is large compared to the (unknown) MHT at 65°N, then setting the MHT at 65°N to zero introduces only a small relative error. This is what we mean by saying the MHT "will be accurate at any latitude where the MHT is large compared to the MHT at 65°N."

This approximation can be quantified at 26ºN by drawing on direct observations from the RAPID and OSNAP arrays. These show that the variability of MHT at 26ºN (from RAPID) is more than four times larger than at 60ºN (from OSNAP), suggesting that at lower latitudes the contribution from the HTCs dominates the total MHT signal. Consequently, any uncertainty introduced by assuming zero transport at 65ºN becomes negligible at latitudes like 26ºN where the integrated HTC is large.

Line 524: 'four time larger at RAPID than at OSNAP' – are the comparisons only based on the MHT estimates from this study?

No, the comparison is not based on the MHT estimates from our study. It relies on independent observational estimates derived from the RAPID and OSNAP arrays, as noted in the original submission where we wrote: "estimates of MHT from the RAPID and OSNAP arrays." These observational datasets provide a direct and independent basis for comparing the variability of MHT at 26°N and 60°N.

We have slightly reworded the original text to make it clearer that the comparison is based on independent observations.

Line 529: What kind of error is this referring to? The mean value of MHT1 is not supposed to affect the derived variations.

We appreciate the reviewer's comment and agree that the time-mean value of $MHT_1$ does not affect the variability of MHT at other latitudes. However, it does influence the time-mean MHT at those latitudes. As stated in the manuscript, the MHT at any latitude is calculated as the sum of $MHT_1$ and the integrated HTCs between 65°N and that latitude. Thus, omitting the time-mean of $MHT_1$ introduces a bias in the resulting time-mean MHT values elsewhere.

From the RAPID and OSNAP arrays, we know that the time-mean MHT is approximately 1.19 PW at 26°N (RAPID) and 0.51 PW at 60°N (OSNAP). Therefore, ignoring the time-mean MHT at 65°N would result in an underestimate of the mean MHT at 26°N by over 40%. In contrast, the variability of MHT at 26°N is more than four times larger than at 60°N, so the influence of $MHT_1$ on the variability is minimal.

To account for this, we set the time-mean MHT at 60°N equal to the mean value from the OSNAP-derived MHT.

Line 532: How representative is the 2014-2018 mean? over a longer period 2004-2020?

The time-mean MHT from the RAPID array is 1.19 PW for the full period 2004–2020, and 1.21 PW for the shorter period 2014–2018. The difference between these two estimates (0.02 PW) is an order of magnitude smaller than the associated uncertainty (approximately 0.2 PW), indicating that the shorter period mean is representative of the longer-term mean.

At 60°N, the OSNAP-derived time-mean MHT is expected to be even more robust, given that the variability is smaller relative to the mean. Specifically, the standard deviation of the OSNAP time series is only 0.03 PW, while the mean is 0.51 PW, further supporting the stability of the time-mean estimate over time.

Line 536: I am not sure about this assumption that is based on a 4-year time series.

We agree with the reviewer that this assumption might not hold exactly, but there is no definitive way to verify the validity of the assumption. That is why we highlight this as a potential limitation of the estimates in the Discussion. Specifically, we wrote:

> "*it is important to remember that the time-mean MHT at 60° N has been set equal to that derived from the OSNAP array over the period 2014-2018. This assumes a constant time-mean MHT at 60° N over the analysis period (2004-2020), and thus deviations from this premise will introduce an error into the time-mean MHT at all the other latitudes*"

Line 540-546: I found the explanations inadequate – this is related to my main comment above. First, the observed MHT from RAPID is most likely the best estimate one can get, so why does depending on it become an issue? I cannot follow the reasoning behind the second point. Why does the observed MHT from RAPID introduce large errors? It is understandable that the RAPID data may be used to first validate this method. But after that validation, could and should it be used to improve the estimates?

We refer the reviewer to our response to a similar comment earlier in this letter. As explained there, we prefer not to publish the MHT estimates constrained by the RAPID-derived MHT. However, if the reviewer feels strongly that these estimates should be included, we would be happy to make them available on Zenodo alongside the other estimates.

Line 549: Once again, it is unclear why three surface heat flux (Qsfc) datasets are used separately, which are over different time periods.

This is no longer the case, as explained earlier in this response letter.

Line 573: Please justify 'very significant' – what is p-value?

Thank you for the comment. We would like to clarify that the MHT time series in our analysis are derived from a Bayesian posterior distribution. In this context, the concept of a frequentist p-value is not directly applicable, since the Bayesian framework does not involve hypothesis testing in the frequentist sense. However, we agree that "very significant" reads vague and, therefore, we have reworded this to "statistically significant at the 95% confidence level".

Line 575: Why is a discrepancy only occurring in 2020?

We did not mean to suggest that a discrepancy occurred only in 2020. As stated in line 575 of the original manuscript, "*The lower correlation observed during the longer period of 2004–2020 is primarily due to a discrepancy in 2020.*" That is, the difference in correlation between the shorter and longer periods is largely driven by the mismatch in 2020, not that discrepancies are entirely absent in other years. In any case, this paragraph has been revised and we hope it is clearer now what we meant.

Line 577: 'This discrepancy is … entire period.' Hard to understand what it means – please rephrase.

This paragraph has been partly rewritten in the revised paper.

Line 588: Figure 4: How are the CIs determined? It is worth a dedicated subsection in Methods on uncertainty in the MHT estimates.

The CIs are computed as the 5th to 95th percentiles of the posterior distribution, based on samples obtained from the BHM. Since the calculation is straightforward once the posterior samples are available, we did not initially include a dedicated explanation. However, in response to the reviewer's comment, we have added the following clarification at the beginning of Section 5.2:

> "*The CIs are computed as the 5th to 95th percentiles of the posterior distribution, based on samples obtained from the BHM.*"

Line 599: As mentioned above, is the difference in the mean MHT between BMH solutions mostly related to Qsfc?

As noted earlier in this letter, we now focus exclusively on the solution derived from the new surface heat flux product, so the comparison between earlier BHM solutions is no longer relevant to the manuscript. However, to answer the reviewer's question: yes, the differences in mean MHT between the previous BHM solutions were primarily due to the use of different surface heat flux products, which was the only distinction between those solutions.

Line 600: 'To complete our comparison' may not be a good motivation. E.g., why apply 5-quarter running averages? How does it help complete the comparison, or how does it help understand the discrepancies?

Thank you for the comment. We have reworded the sentence in the revised manuscript for greater clarity. Our intent in applying a running mean is to extend the comparison across different temporal scales. Agreement at the quarterly scale does not necessarily imply agreement at lower frequencies, and smoothing helps reveal whether discrepancies persist (or are reduced) at longer timescales. This complements the analysis of high-frequency variability and provides a more complete picture of agreement between our estimate and the one from RAPID. In response to an earlier suggestion by the

reviewer, we now use a 4-quarter running mean instead of the original 5-quarter window.

Line 609: The data are 5-year averages. What do the differences during 2005-2007 represent?

There seems to be some confusion; our analysis is based on 5-quarter running averages (4-quarter running averages in the revised manuscript), which are effectively 1-year averages. Therefore, the differences observed during 2005–2007 reflect variability at approximately annual timescales, not 5-year averages.

Line 626: For the comparing purposes, why not apply the same 12-month (4-quarter) running averages to the MHT estimates from this study? That would help make meaningful comparisons.

In response to a comment by reviewer #2, we no longer show time series from Trenberth et al. (2019). Note, however, that we now use a 4-quarter running mean instead of the original 5-quarter window.

Line 628: 'several interesting features' reads odd.

This sentence has been reworded. Thank you.

Line 628 and the whole paragraph: Those features are related to the similarities and differences between BHM solutions. But it is not clear what we will gain from those comparisons. Please refer to my main comments.

Thank you for the comment. Please note that in the revised manuscript, we no longer include the comparison between Bayesian solutions derived from different surface heat flux products, as our focus is now on the solution based on the new observation-based product.

Line 658: As mentioned earlier, would it be better to use the 12-month smoothed data when comparing with Trenberth et al. (2019)?

As mentioned earlier, we no longer show time series from Trenberth et al. (2019). Note, however, that we now use a 4-quarter running mean instead of the original 5-quarter window.

Line 667: The mean is obtained over different lengths of record and different periods. Given the strong interannual and decadal variations in the OHC and probably in the MHT, the time-mean estimates could be biased and cannot be compared directly to each other. Please justify the choices of those estimates to compare with and discuss the comparisons to avoid misinterpretation.

We should mention first that, in the revised paper, we have removed the estimate from Ganachaud & Wunsch (2003), as this was for a completely different period, and have added an estimate from a more recent paper (Liu et al., 2022).

We agree with the reviewer that differences in the period might explain some of the discrepancies between the BHM estimates and those from other studies. To evaluate this, we have calculated the time-mean MHT from the BHM over the Liu22 period (2006-2013) and compared it to BHM estimates for the period 2004-2020. We have found that the differences are less than 10% of the time-mean MHT at all latitudes, suggesting that the discrepancies with T19 and Liu22 are unlikely to be explained by differences in the time period alone. We have added the results of this analysis to the revised manuscript.

Line 686: Is it because of a similar method (MHT as a residual from heat budgets)?

It is difficult to say with certainty. While both our study and T19 derive MHT as a residual from heat budgets, the approaches differ significantly in how the data sets are combined and how the budgets are evaluated. In addition, the specific data sets used in T19 differ from those used in our study. Therefore, the agreement may be due to a combination of methodological and data-related factors rather than from a shared approach alone. We have added a sentence along these lines to the revised manuscript.

Line 688: Why compare to GW03? What can we learn from this comparison?

As mentioned above, we have removed GW03 from the comparison.

Line 706: What is the main objective of this study? A data set (MHT estimates) or a valid method? Please refer to my main comments.

The main goal of this study is to provide new observation-based estimates of MHT across most of the Atlantic Ocean using a statistically rigorous method that integrates data from multiple sources within a Bayesian hierarchical modelling framework. While the methodology is important, particularly the way it integrates multi-source data and propagates uncertainty through joint spatiotemporal modelling, the primary aim is to generate reliable, probabilistic MHT estimates over a broad range of latitudes. These estimates are intended to complement direct observational records (e.g., from the RAPID array), extending our spatial and temporal coverage. We believe this goal is clearly stated in the Introduction of the manuscript, where we motivate the need for spatially and temporally continuous MHT estimates and describe the advantages of our approach in achieving this goal.

Line 710: It is not clear why three solutions are needed.

Please note that, as mentioned earlier in this letter, we no longer show three Bayesian solutions.

Line 719: This seems to be a hasty conclusion. Those correlations are based on data for different time periods and are based on different assumptions.

We thank the reviewer for this comment. In the revised manuscript, all correlation values are computed over the same time periods as those reported in the previous studies, allowing for a fair comparison. We agree that the estimates are based on different assumptions and methodologies, but that is precisely the point of the comparison. We aim to assess how well our BHM-derived estimates align with RAPID-derived MHT relative to earlier approaches, which necessarily involves comparing methods built on different assumptions and approaches. This comparison is important to demonstrate the added value of accounting for spatiotemporal dependencies and properly propagating uncertainty when combining hydrographic and satellite data. We have revised the paragraph to better emphasize this point.

Line 726: It simply indicates that surface heat flux is a major source of uncertainty in the MHT estimates. Please refer to my main comments.

We agree with the reviewer. Since we are no longer comparing solutions based on different surface heat flux products, this sentence has been removed from the manuscript.

---

## Author Comment (AC2)

**Response letter**

We thank the reviewers for their valuable and constructive comments. The reviewers have raised important points that provide us with an opportunity to clarify a number of aspects relating to our results and the Bayesian model. In the following, we explain how the manuscript has been changed and provide a point-by-point response to each of the reviewers' comments. For clarity, our responses are in standard font whereas any text from the reviews is denoted by blue font. Text from the manuscript is italicized.

The main changes made to the manuscript are outlined below:

1. We use a new observation-based surface heat flux data set and focus our analysis on the Bayesian estimates derived from this new data set, in response to a comment by Reviewer #2. Accordingly, we no longer include Bayesian estimates based on ERA5 and NCEP data.
2. We now account for errors in the ocean mass from GRACE due to the GIA correction, geocentre motion and Earth oblateness.
3. We provide a clearer justification for our assumption of zero MHT at 65ºN, in response to a comment by Reviewer #1.

**Referee #2:**

This study tackles the challenge of estimating variations in Atlantic meridional heat transport (MHT) using satellite data and in situ temperature and salinity profiles across 12 latitudinal cross-sections. The authors propose a method to estimate MHT by combining changes in ocean heat content (OHC) with surface heat flux (HF) data. Recognizing the limitations imposed by sparse hydrographic observations, the study advances traditional methods by integrating hydrographic data with satellite altimetry and gravimetry within a joint spatiotemporal Bayesian framework. This fusion enables the generation of probabilistic MHT estimates from 2004 to 2020 across 12 Atlantic latitudinal sections, from 65°N to 35°S. The methodology effectively leverages the comprehensive spatial coverage of satellite data to compensate for the uneven distribution of in situ observations, thereby improving the quality of MHT estimates. Validation against independent measurements from the RAPID array at 26°N (which were not used in the derivation of the MHT estimates) shows good agreement in both the magnitude and timing of variability (with correlations of 0.77 for the raw series and 0.93 for the smoothed series), as well as in the mean transport value (1.17 PW). Results at other latitudes are consistent with prior estimates.

This work addresses the critical issue of variability in Atlantic meridional heat transport—a key component of global and regional ocean heat transport influencing climate. Continuous and precise measurements of Atlantic MHT are essential but limited by the high costs and logistical challenges associated with direct ocean observation systems, which currently provide data at only a few latitudes (e.g., through the RAPID and OSNAP arrays). The authors manage to overcome this limitation here by infering ocean heat transport convergence (HTC) as a residual from the imbalance between OHC changes and surface heat fluxes, using all available data to estimate OHC (i.e., satellite altimetry, gravimetry, and hydrographic observations) within a Bayesian framework. As such, this study is original and highly relevant to the climate science community. The ocean energy budget approach used to derive HTC is not new, nor is the combination of satellite altimetry, gravimetry, and hydrography to estimate OHC. The novelty of this study lies in its application of a Bayesian statistical framework that explicitly accounts for uncertainties in each dataset.

While the overall approach is sound and the results are promising, there are several significant limitations in the current version of the manuscript that must be addressed for the study to be fully convincing.

We thank the reviewer for their valuable and constructive feedback, which has helped us to improve the manuscript.

The reviewer writes that "the novelty of this study lies in its application of a Bayesian statistical framework that explicitly accounts for uncertainties in each dataset.". We appreciate the reviewer's recognition of the Bayesian framework used in our study, but we respectfully note that this comment does not fully capture the novelty of our approach. The core innovation lies not only in accounting for uncertainty in each dataset but in how we integrate multiple data sources (hydrographic, altimetric, and gravimetric) within a joint spatiotemporal Bayesian hierarchical model. This framework allows us to account not just for individual uncertainties but also for spatial and temporal dependencies within and across variables, something that previous studies focused on ocean heat content have not done.

Earlier approaches have typically combined hydrographic and satellite data in a pointwise manner, treating each dataset independently and often ignoring both error structures and spatial correlations. By contrast, our approach enables sharing of information across space, time, and data types in a statistically rigorous way. This leads to more robust and spatially coherent estimates of ocean heat content, heat transport convergence, and ultimately meridional heat transport, with improved quantification of uncertainty.

Major Concerns

Surface Heat Flux (HF) Datasets:

The datasets used to estimate HF are not state-of-the-art. The authors rely on outputs from atmospheric reanalyses, whose surface flux estimates are known to suffer from inconsistencies and large biases due to the weak observational constraints on the short-term forecasts used to generate them. A more robust alternative involves estimating surface fluxes from the atmospheric energy budget using CERES observations at the top of the atmosphere (TOA) and computing the divergence of atmospheric energy transport from reanalysis fields (e.g., winds and temperature), which are more strongly constrained through data assimilation than the short-term forecasts. This approach has been adopted by Mayer et al. (2017, 2021, 2022, 2024) and Meyssignac et al. (2024), and is now widely accepted as yielding net surface fluxes with smaller large-scale biases than reanalysis output-based or satellite-derived model outputs. The authors are strongly encouraged to apply this method, which would substantially increase the reliability of their MHT estimates.

We thank the reviewer for this comment. We would like to note first that, to the best of our knowledge, the DEEP-C product that we used in the original submission was derived using the approach that the reviewer recommends. However, DEEP-C stops in 2017, which is not ideal. For this reason, and following the reviewer's comment, we are now using a new surface heat flux product derived using the approach recommended by the reviewer. Specifically, the net surface HF is calculated by combining top-of-the-atmosphere (TOA) radiative flux with the divergence of vertical integral of total energy flux and the tendency of vertical integral of total energy as described in Mayer et al. (2022). The TOA flux has been obtained from the Clouds and the Earth's Radiant Energy System–Energy Balanced and Filled (CERES-EBAF) Edition-4.2.1 monthly data product (Loeb et al., 2018), whereas the vertically integrated atmospheric energy quantities are those of Mayer et al. (2022).

Note also, as mentioned earlier in this response letter, that the revised manuscript focuses on estimates of MHT derived using this new surface heat flux product. That is, we no longer include Bayesian estimates based on ERA5 and NCEP data. The ERA5 and NCEP heat flux products are used only to obtain a measure of uncertainty in the surface heat flux data; they are not used directly in the BHM.

Uncertainty Estimation in GRACE Data:

Given the central role of uncertainty quantification in this study, it is concerning that the uncertainty associated with space gravimetry data is only partially addressed. The authors rely on uncertainties from the mascon product, which do not account for critical error sources in GRACE data, such as the glacial isostatic adjustment (GIA), geocenter motion, and C20 corrections. These components are known to dominate the error budget in ocean mass estimates from GRACE (see Quinn & Ponte 2010; Blazquez et al. 2018; Uebbing et al. 2019) and significantly impact thermal expansion

estimates. The authors should incorporate these additional uncertainty sources into their analysis.

We thank the reviewer for this comment. As mentioned at the beginning of this letter, we now account for errors in the ocean mass from GRACE due to the GIA correction, geocentre motion and Earth oblateness.

Thermosteric Sea Level (TS) and OHC Relationship:

The assumed relationship between thermosteric sea level and vertically integrated ocean heat content in the process layer involves important simplifications. Specifically, the neglect of the deep ocean (below 1500 m) and the linearity assumption between TS and vertically integrated OHC (see Eq. 17) could introduce significant inconsistencies. These approximations should be explicitly discussed and, if possible, their impact quantified.

Thank you for this comment. First, we would like to clarify that the contribution from the deep ocean (below 1500 m) to thermosteric sea level (TS) is not neglected in our analysis. As described in the manuscript, we account for the full-depth contribution through the relationship SL = TS + HS + OM, where the sea level (SL) from satellite altimetry reflects the total steric signal, including from the deep ocean. We accommodate the potential deep-ocean contribution by inflating the uncertainties in the TS and HS data by 20%. This allows the Bayesian hierarchical model to reconcile the observed altimetric sea level with full-depth TS and HS contributions, including those below 1500 m.

Regarding the second point raised by the reviewer, we agree that, because the thermal expansion coefficient ($\alpha$) varies with depth, assuming a constant $\alpha$ to relate vertically integrated heat transport convergence to changes in TS is an approximation. However, this simplification is necessary because modelling the relationship exactly would require knowledge of the vertical structure of the heat transport convergence (or the velocity and temperature fields), which is not available. Moreover, our Bayesian framework operates on two-dimensional, vertically integrated fields; incorporating vertical variations into the framework would significantly increase the model's complexity and is beyond the scope of this study.

We tested the sensitivity of our estimates to the depth over which $\alpha$ is averaged and found the results to be relatively insensitive. While this test does not fully rule out the possibility of small biases, it gives us confidence that the approximation is unlikely to introduce substantial errors.

In response to the reviewer's comment, we have added the following paragraph:

> *"Note that Eq. (16) assumes a constant α to relate vertically integrated HTC to changes in TS. While this neglects vertical variations in α, the approximation is necessary because modelling the relationship exactly would require knowledge of the vertical structure of HTC, which is not available. Furthermore, our Bayesian framework operates on two-dimensional, vertically integrated fields; incorporating vertical variations into the framework would significantly increase the model's complexity. Nevertheless, to assess the impact of this assumption, we tested the sensitivity of our estimates to different choices of α, obtained by averaging over different depth ranges. The results show almost no effect on the phase of the estimated HTC variability and only a small effect on its amplitude. While this test does not fully rule out the possibility of small biases, it gives us confidence that the approximation is unlikely to introduce substantial errors."*

We have also added the following sentence to the Conclusions:

> *"Note also that our Bayesian model operates in two dimensions and therefore uses a vertically integrated thermal expansion coefficient to relate changes in TS to the vertically integrated HTC, which may introduce some approximation error."*

Comparison with Outdated MHT Estimates:

The authors compare their results with an outdated MHT estimate based on ERA-Interim and CERES surface fluxes. ERA-Interim, in particular, is known to suffer from a negative radiation

budget at TOA, which inevitably biases surface flux estimates. More recent and accurate estimates using ERA5 are available (e.g., Meyssignac et al. 2024; Mayer et al. 2022; Liu et al. 2020). The authors should compare their results against one of these more recent and reliable datasets.

We thank the reviewer for this helpful suggestion regarding more recent MHT estimates. In response, we have removed the estimate from Ganachaud & Wunsch (2003), as it corresponds to a different period and so is less relevant for comparison. We have also included the time-mean MHT estimates from Liu et al. (2022), which were kindly provided by Chunlei Liu upon request. Unfortunately, the full time series were not available for comparison.

Regarding the other two studies mentioned by the reviewer (Meyssignac et al., 2024; Mayer et al., 2022), we note that they obtained MHT estimates only at 26°N. At this latitude, we already compare our estimates to the RAPID array observations, which offer a more direct and independently measured benchmark. Additionally, to the best of our knowledge, the MHT estimates from these two studies are not publicly available.

Detailed Comments

L204: Is the GIA correction used in GRACE consistent with that used in the altimetry analysis?

We appreciate the reviewer's attention to this detail. In our analysis, the GIA corrections applied to the GRACE and altimetry data do not come from the same model. However, we explicitly account for uncertainty in the GIA correction in both cases. Specifically, uncertainty in the GIA correction is incorporated into the error structure of the GRACE and altimetry observations within the Bayesian framework.

Moreover, the GIA correction applied to the altimetry data is relatively small, with a domain-averaged absolute value of approximately 0.3 mm/yr and a range from -0.57 to 0.37 mm/yr across the domain. Given this, we expect the use of different GIA models to have a minimal effect on our results.

L205: The GRACE error budget is dominated by uncertainties in GIA and geocenter corrections, which are not currently accounted for in the analysis (see major concern #2).

We are now accounting for these error sources. Please see our response to major concern #2.

L217: surface fluxes derived from the output of reanalyses are biased. Use instead a combination of CERES TOA data and vertically integrated atmospheric energy divergence estimated from reanalyses, as in Mayer et al. (2022). (See major concern #1.)

We are now using the surface heat flux recommended by the reviewer. Please see our response to major concern #1.

L240: For the effective resolution of satellite altimetry, refer to the updated analysis in Ballarotta et al. (2019).

Thank you for this reference, which has been added to the revised manuscript.

L271: More satellites are currently operating with ~30-day repeat cycles (e.g., Sentinel-3A/B, AltiKa, CryoSat) than with 10-day cycles. The term "most" should be removed or corrected.

We agree with the reviewer, but the sentence is correct as written. The sentence in question does not state that most satellites have a 10-day repeat cycle, but rather that most operate with a repeat cycle in the range of 10 to 35 days.

L333: Clarify the term "reference datum." If referring to the altimetry reference ellipsoid, note that all altimetry satellites use the same reference ellipsoid. Biases across satellite altimeters' data arise rather because of biases in the radar signal characterization (e.g. biases in the radar signal delay characterisation, biases in the antenna center positioning relative to the center of mass of the satellite, etc…).

Thank you for this comment. To clarify, when we refer to differences in "vertical datums" we are not

referring to inconsistencies between altimetry missions, which indeed use the same reference ellipsoid. Rather, we are referring to potential vertical offsets between the different data sources used in our analysis, specifically thermosteric sea level, halosteric sea level, GRACE-derived ocean mass, and satellite altimetry. Our approach accounts for these potential offsets by including a constant bias term in the Bayesian model.

In response to the reviewer's comment, the sentence in question has been reworded for clarity.

L396: Is c(Rj) computed over the full water column or limited to 0–1500 m? How is the deep ocean contribution addressed? This is important, especially since changes in deep ocean heat content can affect the α–C relationship not only on the time-mean but also over time. (See major concern #3.)

Both $\alpha$ and $c$ are averaged over the top 1500 m of the ocean, although we tested the sensitivity of our estimates to different choices of $\alpha$ and $c$, obtained by averaging over different depth ranges. Please see our response to major concern #3.

L598: BHM2 is likely biased low due to the negative TOA radiation budget in ERA-Interim. Why compare to such an outdated estimate (Trenberth et al. 2019) instead of more recent ones using ERA5? (See major concern #4.)

As mentioned earlier in this response letter, we no longer include estimates derived from ERA5 or NCEP. We have added a more recent estimate to the comparison, as requested by the reviewer. Please see our response to major concern #4.

References:

Ballarotta, M., Ubelmann, C., Pujol, M.-I., Taburet, G., Fournier, F., Legeais, J.-F., Faugère, Y., Delepoulle, A., Chelton, D., Dibarboure, G., and Picot, N.: On the resolutions of ocean altimetry maps, Ocean Sci., 15, 1091–1109, https://doi.org/10.5194/os-15-1091-2019, 2019.

A Blazquez, B Meyssignac, JM Lemoine, E Berthier, A Ribes, A Cazenave, Exploring the uncertainty in GRACE estimates of the mass redistributions at the Earth surface: implications for the global water and sea level budgets, Geophysical Journal International, Volume 215, Issue 1, October 2018, Pages 415–430, https://doi.org/10.1093/gji/ggy293

Liu C et al (2020) Variability in the global energy budget and transports 1985-2017. Climate Dyn 55:3381–3396. https://doi.org/10.1007/s00382-020-05451-8

Mayer M, Haimberger L, Edwards JM, Hyder P (2017) Toward consistent diagnostics of the coupled atmosphere and ocean energy budgets. J Climate 30:9225–9246. https://doi.org/10.1175/JCLI-D-17-0137.1

Mayer J, Mayer M, Haimberger L (2021) Mass-consistent atmospheric energy and moisture budget monthly data from 1979 to present derived from ERA5 reanalysis. Copernic Clim Change Serv (C3S) Clim Data Store (CDS). https://doi.org/10.24381/cds.c2451f6b

Mayer J, Mayer M, Haimberger L, Liu C (2022) Comparison of surface energy fluxes from global to local scale. J Clim. https://doi.org/10.1175/JCLI-D-21-0598.1

Mayer M, Kato S, Bosilovich M et al (2024) Assessment of atmospheric and surface energy budgets using observation-based data products. Surv Geophys. https://doiorg.insu.bib.cnrs.fr/10.1007/ s10712-024-09827-x

Meyssignac, B., Fourest, S., Mayer, M. et al. North Atlantic Heat Transport Convergence Derived from a Regional Energy Budget Using Different Ocean Heat Content Estimates. Surv Geophys 45, 1855–1874 (2024). doi 10.1007/s10712-024-09865-5

Katherine J. Quinn, Rui M. Ponte, Uncertainty in ocean mass trends from GRACE, Geophysical Journal International, Volume 181, Issue 2, May 2010, Pages 762–768, https://doi.org/10.1111/j.1365-246X.2010.04508.x

Trenberth, K. E., Y. Zhang, J. T. Fasullo, and L. Cheng, 2019: Observation-Based Estimates of Global and Basin Ocean Meridional Heat Transport Time Series. J. Climate, 32, 4567–4583, https://doi.org/10.1175/JCLI-D-18-0872.1.

Uebbing, B., Kusche, J., Rietbroek, R., & Landerer, F. W. (2019). Processing choices affect ocean mass estimates from GRACE. Journal of Geophysical Research: Oceans, 124, 1029–1044. doi 10.1029/2018JC014341

---

## Author Response (AR2)

**Response letter**

After this second round of reviews, our paper has been accepted for publication as is, and therefore a response letter is not required. Nevertheless, we would like to thank the new reviewer (Referee #3), who was assigned to our paper after the first round of reviews, for their valuable comments.